# Retrieval-Augmented Perception:
# High-Resolution Image Perception Meets Visual RAG

Wenbin Wang [1]   Yongcheng Jing [2]   Liang Ding [3]   Yingjie Wang [2]
Li Shen [4]   Yong Luo [1]   Bo Du [1]   Dacheng Tao [2]

## Abstract

High-resolution (HR) image perception remains a key challenge in multimodal large language models (MLLMs). To drive progress beyond the limits of heuristic methods, this paper advances HR perception capabilities of MLLMs by harnessing cutting-edge long-context techniques such as retrieval-augmented generation (RAG). Towards this end, this paper presents the first study exploring the use of RAG to address HR perception challenges. Specifically, we propose *Retrieval-Augmented Perception (RAP)*, a training-free framework that retrieves and fuses relevant image crops while preserving spatial context using the proposed *Spatial-Awareness Layout*. To accommodate different tasks, the proposed *Retrieved-Exploration Search (RE-Search)* dynamically selects the optimal number of crops based on model confidence and retrieval scores. Experimental results on HR benchmarks demonstrate the significant effectiveness of *RAP*, with LLaVA-v1.5-13B achieving a 43% improvement on $V^*$ Bench and 19% on HR-Bench. Code is available at https://github.com/DreamMr/RAP.

## 1. Introduction

Multimodal large language models (MLLMs) have achieved remarkable progress in vision-language understanding, reasoning, and interaction, leveraging visual signals to process and interpret visual information (Yin et al., 2023). Current

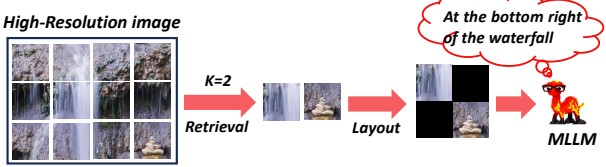

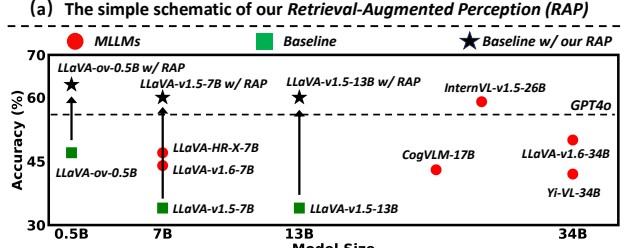

*Figure 1.* (a) Overview of the proposed ***Retrieval-Augmented Perception (RAP)*** framework, which divides the HR images into image crops for retrieval, followed by layout reconstruction to retain the spatial information; (b) Performance comparison of MLLMs across various model sizes, demonstrating consistent improvements with our ***RAP*** on ***HR-Bench***.

MLLMs (Liu et al., 2024a; Bai et al., 2023; Liu et al., 2024b; Wang et al., 2023; Abdin et al., 2024) typically process images at a fixed resolution (*e.g.,* $448 \times 448$). While this design streamlines the computational pipeline, it introduces significant challenges, such as shape distortion and blurring when handling high-resolution (HR) images. These distortions notably impair the performance of MLLMs, especially in tasks that involve analysing real-world images with varying resolutions, such as visual grounding and optical character recognition that demand fine-grained visual details (Zhang et al., 2024a; Tian et al., 2022; 2023; Wang et al., 2024).

In response to this dilemma, emerging research on enhancing the HR image perceptual capabilities of MLLMs has gained increasing attention. Existing approaches can be broadly categorised into three groups: (1) cropping-based methods (Chen et al., 2024c; Liu et al., 2024b; Li et al., 2024c), (2) HR visual encoder methods (Luo et al., 2024; Ge et al., 2024; Lu et al., 2024), and (3) search-based meth-

[1]School of Computer Science, National Engineering Research Center for Multimedia Software and Hubei Key Laboratory of Multimedia and Network Communication Engineering, Wuhan University, Wuhan 430072, China [2]Nanyang Technological University, Singapore 639798 [3]The University of Sydney, Australia [4]Shenzhen Campus of Sun Yat-sen University, China. Correspondence to: Liang Ding <liangding.liam@gmail.com>, Yong Luo <luoyong@whu.edu.cn>, Dacheng Tao <dacheng.tao@gmail.com>.

*Proceedings of the $42^{nd}$ International Conference on Machine Learning*, Vancouver, Canada. PMLR 267, 2025. Copyright 2025 by the author(s).

ods (Wu & Xie, 2024; Wang et al., 2025; Shen et al., 2024). Despite notable progress, both cropping-based and HR visual encoder methods still require downsampling HR images to mitigate excessively long visual token sequences, resulting in substantial loss of fine-grained details. Although search-based methods avoid downsampling, they face several limitations. These methods follow a top-down search from high to low resolution; however, at the initial stage, models struggle to accurately perceive small objects, often resulting in erroneous search paths. Furthermore, search-based approaches rely on hierarchical, layer-by-layer retrieval, preventing parallel processing and rendering them inefficient and cumbersome for deployment.

These limitations prompt our rethinking of the fundamental challenge in HR perception. Ideally, effective HR perception requires an MLLM with robust long-context capabilities— for instance, processing an 8K HR image with ViT-L/14 (Dosovitskiy et al., 2021) generates approximately $\sim$300K visual tokens. This raises the question of whether the key to HR perception lies in enhancing the long-context capacity of MLLMs, rather than relying on existing heuristic approaches, particularly in light of recent encouraging advancements in long-context techniques for general LLMs. In particular, retrieval-augmented generation (RAG) has proven highly effective in recent long-context LLMs, by retrieving crucial fragments and reducing the impact of irrelevant information (Jin et al., 2024). Motivated by this, this paper poses a largely overlooked question: *Is it possible to directly enhance the long-context capability of MLLMs using RAG, as in general LLMs, to overcome the limitations of existing HR perception methods?*

However, exploring this research question presents significant challenges, as images, unlike text, are two-dimensional (excluding the channel dimension) and are characterised by width and height. As a pilot study, we begin by focusing on two key aspects: the layout of retrieved image crops and the impact of the number of retrieved crops on performance. This leads to the following specific challenges: *1) How should the retrieved image crops be organised?* Furthermore, the number of retrieved key fragments plays a critical role in RAG performance (Jin et al., 2024), prompting our second research question: *2) How does the number of retrieved image crops influence the final performance?* Building on insights from these two questions, we further pose a third research question: *3) How can RAG systems be designed to enhance MLLM perception of HR images?*

To address the **1st challenge**, we conduct a series of experiments using the **HR-Bench** (Wang et al., 2025) to investigate the effects of different layout strategies. We evaluate state-of-the-art (SOTA) MLLMs (Liu et al., 2024a;b) across various layout configurations. Specifically, we compare three strategies: 1) arranging the retrieved image crops in

their original order, 2) ordering them in descending order based on retrieval scores (Jin et al., 2024), and 3) preserving the relative positional relationships among the retrieved crops. Our empirical results suggest that maintaining the relative positional relationships of the image crops significantly enhances HR perception, particularly for tasks that depend on spatial relationships.

In response to the **2nd question**, this paper investigates the impact of the number of retrieved image crops. Our findings reveal that the optimal number of retrieved crops depends on the task type. For single-instance perception tasks, a small number of crops suffices for significant performance improvements, whereas too many crops degrade performance due to the high image resolution. In contrast, for cross-instance perception tasks, fewer crops result in information loss and reduced performance, while more crops help preserve essential details and minimise performance degradation. However, an excessive number of crops still harms performance due to challenges from overly high resolution.

In tackling the **3rd question**, we integrate the insights gained from the previous investigations to design a new framework, which we term ***Retrieval-Augmented Perception (RAP)***. As illustrated in Figure 1(a), *RAP* processes high-resolution images by retrieving image crops relevant to the query through VisRAG (Yu et al., 2024). We propose a simple yet efficient layout method, termed as *Spatial-Awareness Layout*, which preserves the original relative spatial relationships among the image crops. To determine the optimal number of retrieved image crops, we introduce a novel scheme termed as *RE-Search* (Retrieved-Exploration Search), which adaptively adjusts the number of crops based on the model's confidence in the sufficiency of the retrieved information.

In particular, VisRAG is first used to compute the similarity scores between each image crop and the query. We then retain the top $K$ crops with the highest similarity scores, ensuring their relative spatial relationships are preserved through the *Spatial-Awareness Layout*. To determine the optimal $K$, we construct a RE-Tree, where each node represents a new image synthesized by retaining different proportions of the image crops. The search process within this tree is guided by both the retrieved similarity scores and the model's confidence in whether the image offers sufficient information to answer the query.

Our contribution is thereby the first investigation into using visual RAG to enhance HR image perception in MLLMs. This is accomplished by a novel *RAP*, a training-free framework that comprises *Spatial-Awareness Layout* to preserve the positions of image crops and *RE-Search* to adaptively select the optimal number of retained crops. Experiments demonstrate that *RAP* consistently delivers significant improvements, with an average accuracy increase of 24% on HR image benchmarks and even general MLLM tasks.

## 2. Related Work

MLLMs consist of a **Visual Encoder** (Dosovitskiy et al., 2021; Radford et al., 2021) for extracting visual features and a **LLM** (Touvron et al., 2023a;b) for decoding text, both initialized from pretrained models. A **multimodal Connector** (*e.g.,* MLP) links the vision and language modalities. To align the resolution used during visual encoder pretraining (*e.g.,* $336 \times 336$ in LLaVA), images are typically resized, which can distort and blur HR images. To address this, existing approaches fall into three categories: **1) cropping-based methods**, **2) HR visual encoder methods**, and **3) search-based methods**.

**Cropping-based methods.** Representative cropping-based methods for HR MLLMs (Chen et al., 2024a; Zhang et al., 2024b; Liu et al., 2024c), such as LLaVA-v1.6 (Liu et al., 2024b) and LLaVA-ov (Li et al., 2024a), segment images into multiple image crops. Each image crop is independently encoded using ViT (Dosovitskiy et al., 2021) and subsequently concatenated for LLM processing.

**HR Visual Encoder.** High-resolution image understanding can be enhanced by incorporating HR visual encoders without substantially increasing the number of visual tokens. For instance, Vary (Wei et al., 2023) and Deepseek-VL (Lu et al., 2024) adopt the SAM (Kirillov et al., 2023) to improve the performance of MLLMs on HR images. MiniGemini-HD (Li et al., 2024b), LLaVA-HR (Luo et al., 2024), and ConvLLaVA (Ge et al., 2024) utilize ConvNeXt (Liu et al., 2022), employing techniques such as cross-attention or adapter to extract visual features.

**Search-based Methods.** Search-based methods organize images into a tree structure to extract query-relevant regions through a top-down approach. $DC^2$ (Wang et al., 2025) leverages visual memory to store objects and coordinates, retrieving crops to generate text and reduce detail loss. Zoom Eye (Shen et al., 2024) employs a tree search algorithm to directly identify and extract relevant crops from HR images. Wu & Xie (2024) propose SEAL, a meta-architecture that actively reasons and retrieves essential visual information.

**Multimodality RAG.** Multimodal RAG tasks include matching images to text and retrieving text-image pairs to answer questions (Chang et al., 2022; Han et al., 2017; Xia et al., 2024a;b). Yu et al. (2024) propose Vision-based Retrieval-augmented Generation to effectively utilize and retain data in multimodal documents.

Existing methods enhance MLLMs' ability to perceive HR images, but processing extremely HR images (*e.g.,* 8K) remains challenging. Inspired by RAG's success in handling long contexts for LLMs, this paper for the first time explores its use to improve MLLMs' HR image perception.

## 3. Pilot Study

In this section, we conduct a systematic investigation into the challenges associated with employing RAG to enhance the perceptual capabilities of MLLMs, motivating the design of the proposed RAP framework in Sect. 4.

### 3.1. Preliminary

In this section, we introduce the pipeline for applying RAG to MLLMs for the perception of HR images. Given an HR image, we divide it into an image crop set, denoted as $V = \{v_1, ..., v_n\}$, where $n$ is the number of image crops. Inspired by Yu et al. (2024), the query and image crops are independently encoded as text and images within the VLM, yielding a sequence of hidden states. Subsequently, the similarity scores between the query embedding and the image crop embeddings are computed. The similarity score $s(q, V)$ is calculated by the cosine similarity of the query and image crop embeddings:

$$s(q, V) = (1 - \frac{q \cdot V^T}{||q|| \cdot ||V||}) \cdot \frac{1}{2}. \tag{1}$$

Finally, the top $K$ image crops are selected based on the $s(q, V)$ to facilitate the MLLM's perception of HR images.

In the following sections, we systematically analyze the impact of retrieved image crop layouts and quantities on **HR-Bench**, which consists of **HR-Bench 8K** and **HR-Bench 4K**. **HR-Bench 8K**, with 8K-resolution images from DIV8K (Gu et al., 2019) and the Internet, includes **Fine-grained Single-instance Perception** (**FSP**) and **Fine-grained Cross-instance Perception** (**FCP**) tasks. Cropping 8K images around relevant objects produces *HR-Bench 4K*.

### 3.2. Impact of the Layout of Retrieved Image Crops

This subsection investigates the relationship between the layout of retrieved image crops and the performance of MLLMs in the RAG system.

**Experimental setting.** We compare three layout strategies: 1) Sort according to the retrieval scores in descending order; 2) After selecting the top $K$ image crops, arrange them in the order in which the image crops appear; 3) Maintain the relative positional relationships of the image crops. We conduct experiments on **HR-Bench** using LLaVA-v1.6-7B.

**Observations.** As shown in Table 1, retrieving key image crops through RAG significantly improves performance on the **FSP** task but results in a noticeable performance drop on the **FCP** task. Furthermore, maintaining the relative positions between each image crop achieves a better performance balance between the **FSP** and **FCP** tasks.

**Insights.** Maintaining the relative positional relationships between retrieved image crops is essential, particularly for

*Table 1.* The effect of different layout strategies. While all three strategies improve fine-grained perception, only strategy 3) excels in **FCP** tasks by preserving positions, achieving superior performance compared to other strategies.

|  | HR-Bench 4K | | | HR-Bench 8K | | |
|---|---|---|---|---|---|---|
|  | *FSP* | *FCP* | *Avg.* | *FSP* | *FCP* | *Avg.* |
| Baseline | 49.0 | **46.8** | 47.9 | 37.2 | **44.2** | 40.8 |
| *+ 1)* | **74.8** | 38.3 | 56.5 | 56.8 | 26.8 | 41.8 |
| *+ 2)* | 72.3 | 38.5 | 55.4 | 61.3 | 25.5 | 43.4 |
| *+ 3)* | 74.0 | 41.5 | **57.8** | 59.5 | 30.0 | **44.8** |

tasks requiring spatial awareness.

### 3.3. Impact of the Number of Retrieved Image Crops

This subsection investigates the relationship between the number of retrieved image crops and the performance of MLLMs in HR image perception.

**Experimental setting.** We analyze the relationship between performance (*i.e.,* accuracy) and the number of the retrieved image crops, using the LLaVA-v1.5 and LLaVA-v1.6.

**Observations.** As shown in Figure 2, we visualize the relationship between the number of retrieved image crops (*i.e.,* $K$) and performance. As $K$ increases, more image crops are introduced, providing additional visual information that enhances performance on **FCP** tasks. However, this also raises the image resolution, increasing the likelihood of the model generating incorrect answers. Conversely, smaller $K$ retains only essential visual information, improving performance on **FSP** tasks but sacrificing significant visual details, which causes a notable performance decline on **FCP** tasks.

**Insights.** Different types of tasks require different numbers of retrieved image crops $K$. For **FSP** tasks, smaller $K$ improves results, but larger $K$ reduces performance by increasing resolution. Conversely, for **FCP** tasks, larger $K$ preserves visual information and outperforms smaller $K$.

## 4. Proposed Retrieval-Augmented Perception

### 4.1. Method Overview

Driven by the aforementioned insights in Sect. 3, we propose a novel framework — **Retrieval-Augmented Perception (RAP)**. The design principle of **RAP** is to retrieve key image crops to replace the original HR image, preserving essential visual information while reducing resolution to improve MLLM perception of HR images. To achieve this, we divide the image into various crops, calculate similarity scores (Eq. 1) with the query, and select the top $K$ image crops to synthesize a new image $V'$. We design a *Spatial-Awareness Layout* algorithm to maintain the relative positional relation-

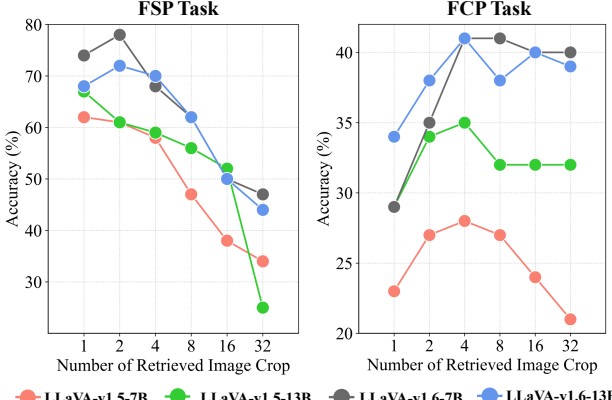

*Figure 2.* The effect of the number of retrieved image crops on model performance. **FSP** and **FCP** represent the fine-grained single-instance perception tasks and fine-grained cross-instance perception tasks, respectively.

ships between the image crops. To adaptively select $K$, we propose *Retrieved-Exploration Search (RE-Search)*, which determines $K$ based on the model's confidence in $V'$ and its similarity to the query. The *Spatial-Awareness Layout* and *RE-Search* are presented in the subsequent sections.

### 4.2. Spatial-Awareness Layout

In Sect. 3.2, we find that maintaing the positional relationship between image crops is essential. Thus, we propose a simple and efficient method, termed *Spatial-Awareness Layout*. We denote $M \in \{0,1\}^{R \times C}$ as a binary matrix of size $R \times C$, where $R$ and $C$ represent the number of rows and columns of image crops $V$, respectively. The $M_{i,j} = 1$ indicates an image crop to be preserved and $M_{i,j} = 0$ indicates the image crops to be removed. We seek to construct a compressed matrix $M'$ by removing any row or column of $M$ that is entirely zero. Formally, we define two index sets:

$$R' = \{i | \exists j \text{ s.t.} M_{i,j} = 1\}, C' = \{j | \exists i \text{ s.t.} M_{i,j} = 1\}. \quad (2)$$

The compressed matrix $M' \in \{0,1\}^{N_r \times N_c}$, with $N_r = |R'|$ and $N_c = |C'|$, is then constructed according to: $M'_{\tilde{i},\tilde{j}} = M_{i,j}$, where $i = R'[\tilde{i}]$ and $j = C'[\tilde{j}]$. This guarantees that $M'$ retains all rows and columns of $M$ containing at least one entry equal to $1$, effectively discarding rows and columns composed entirely of zeros. Moreover, an mapping function $\Phi : \{0,...,N_r-1\} \times \{0,...,N_c-1\} \to \{0,...,R-1\} \times \{0,...,C-1\}$ is defined as $\Phi(\tilde{i},\tilde{j}) = (R'[\tilde{i}], C'[\tilde{j}])$, thereby enabling each coordinate $(\tilde{i},\tilde{j})$ in the compressed matrix $M'$ to be mapped back to its original position $(i,j)$ in $M$. Finally, we initializes an blank image $V'$ and iterates over the mapping $\Phi$, where each pair $(\tilde{i},\tilde{j})$ is mapped to $(i,j)$. For each mapping, $V[i][j]$ is assigned to the corresponding $V'[\tilde{i}][\tilde{j}]$. We use image $V'$ to replace the HR image $V$ for the MLLM to answer the query. The implement

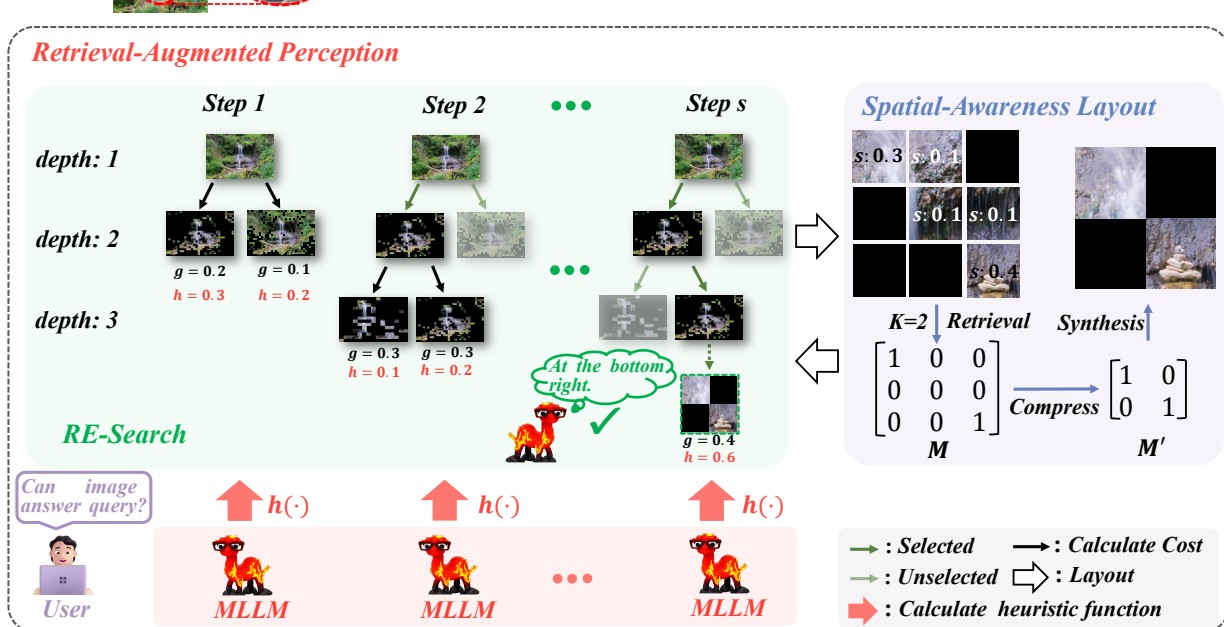

*Figure 3.* Detailed illustration of our proposed **RAP** with a running example. We firstly divide HR image into multiple image crops and compute the similarity score $s$ between the query and image corps to retrieve the key image crops. We design a simple and efficient method called **Spatial-Awareness Layout** to maintain the relative positional relationships of the image crops. Since the number of image crops is highly sensitive to the task type, we propose **RE-Search**, which identifies the optimal $K$ based on the model's confidence scores and retrieval scores.

of *Spatial-Awareness Layout* is shown in Algorithm 1.

### 4.3. Retrieved-Exploration Search

In Sect. 3.3, we find that different types of tasks significantly influence the choice of $K$. Here, we utilize a search algorithm to obtain the optimal $K$. For search algorithm, we consider two primary factors: ***Efficiency***, ensuring high efficiency for optimal user experience, and ***Robustness***, guaranteeing consistent results across multiple runs in image perception tasks. Existing tree-search methods (Wang et al., 2025) require visiting all nodes, leading to low efficiency. With the development of O1, many recent works (Yao et al., 2024; Zhao et al.) employ Monte Carlo Tree Search (MCTS) to find the optimal reasoning path. However, MCTS relies on random sampling, resulting in a lower robustness. $A^*$ search algorithm uses a heuristic function to intelligently guide its exploration. This heuristic allows $A^*$ to prioritize promising paths, significantly accelerating the search process. Furthermore, $A^*$ explores the nodes in the same order and find the same optimal path, ensures high robustness. However, effectively defining the state representation and designing an appropriate heuristic function for $A^*$ is a non-trivial challenge.

Building upon the strengths of $A^*$, we introduce *Retrieved-Exploration Search (RE-Search)*. In the following parts, we will elucidate the RE-Tree, a novel structure that elegantly represents the search states within *RE-Search*, and the REward function, which serves as the guiding heuristic for this innovative approach.

**RE-Tree Representation.** Inspired by Wang et al. (2025); Shen et al. (2024), we model the HR image as a tree. Unlike existing search-based methods, we represent distinct nodes at the same layer by preserving different $K$ image crops. This enables the model to perceive lower-resolution images from the begining, mitigating the risk of the MLLM converging to suboptimal solutions. We denote $P = \{p_1, ..., p_n\}$ as the retention ratio. For instance, for the first child node $n_1$, we retain the top $N' \times p_1$ image crops. The $N'$ represents the number of image crops for the current image. To obtain a complete image for calculating the REward function, we employ *Spatial-Awareness Layout* to assemble the individual image crops into a complete image $V'$.

**REward Function.** $A^*$ search is a best-first search algorithm that prioritizes nodes with the lowest combined cost, calculated as the sum of the actual cost $g(t_s)$ from the start node $t_0$ to $t_s$ and the estimated cost $h(t_s)$ to the goal. In our *RE-Search*, the path from $t_0$ to $t_s$ is represented as the

**Algorithm 1** *Spatial-Awareness Layout*

**function** $SpatialLayout(V, M)$
  $R' \leftarrow \{i \mid \exists j \text{ s.t.} M_{i,j} = 1\}$
  $C' \leftarrow \{j \mid \exists i \text{ s.t.} M_{i,j} = 1\}$
  $N_r \leftarrow |R'|, \quad N_c \leftarrow |C'|$
  Construct a binary matrix $M' \in \{0,1\}^{N_r \times N_c}$
  **for** $\tilde{i} = 1 \rightarrow N_r - 1$ **do**
    **for** $\tilde{j} = 0 \rightarrow N_c - 1$ **do**
      $i \leftarrow R'[\tilde{i}], \quad j \leftarrow C'[\tilde{j}]$
      $M'_{\tilde{i},\tilde{j}} \leftarrow M_{i,j}$
    **end for**
  **end for**
  Initialize a blank image $V'$
  **for** $\tilde{i} = 0 \rightarrow N_r - 1$ **do**
    **for** $\tilde{j} = 0 \rightarrow N_c - 1$ **do**
      $i \leftarrow R'[\tilde{i}], \quad j \leftarrow C'[\tilde{j}]$
      **if** $M'_{\tilde{i},\tilde{j}} = 1$ **then**
        $V'[\tilde{i},\tilde{j}] \leftarrow V[i,j]$
      **end if**
    **end for**
  **end for**
  return $V'$
**end function**

progression from the original HR image to the currently retained top-$K$ image crops. We use the similarity score between these $K$ image crops and the query as $g(t_s)$:

$$g(t_s) = \frac{1}{n} \sum_{i=1}^{n} s(q, v_i), \tag{3}$$

where $n$ represents the number of image crops, and $v_i$ represents the $i$-th image crops for current image $V$. Inspired by Shen et al. (2024), we use the model's confidence in whether the current image $V$ can answer the given query as the cost from $t_s$ to the goal:

$$h(t_s) = 1 - \mathcal{P}_\theta(\text{"Yes"}|p_h(q), V), \tag{4}$$

where $\mathcal{P}_\theta$ represents the MLLM and $p_h(\cdot)$ represents the prompt (*e.g., "Question: {q}. Could you answer the question based on the available visual information? Answer Yes or No."*) used to query the MLLM for calculating the confidence that the answer is "Yes". We utilize the model's confidence to estimate the cost from the current to the target node, analogous to the heuristic function in the $A^*$ algorithm. A lower $h(\cdot)$ indicates a higher likelihood of containing essential information, warranting prioritized exploration.

Since MLLM cannot accurately perceive the HR image at the beginning, the $h(t_s)$ provided at shallow depths of the tree is unreliable. As the tree depth increases and the image resolution gradually decreases, the model becomes more confident in determining whether the current image can

answer the query. Therefore, we assign a lower weight to $h(t_s)$ at the beginning and gradually increase its weight as the tree depth grows. Mathematically, the cost function $f(t)$ can be written as:

$$f(t_s) = (1 - w) \cdot g(t_s) + w \cdot h(t_s), \tag{5}$$

$$w = (1 - b) \cdot (1 - \frac{1}{d})^2 + b, \tag{6}$$

where $b$ is a bias value, set here at 0.2 and $d$ denotes the depth of the image tree.

### 4.4. Algorithmic Workflow

In this section, we introduce how to use our ***RAP*** to perceive HR image. Given a HR image $I$, we first divide the HR image into various image crops $V$, with the size of each image crop not exceeding the predefined resolution of the retriever's image encoder. Subsequently, we utilize VisRAG (Yu et al., 2024) to compute the cosine similarity between the query and image crops. We use *RE-Search* to search the optimal $K$ image crops and using the *Spatial-Awareness Layout* to synthesize the image $V_f$, which replaces the original HR image $V$ as input to the MLLM. We denote $c$ as the answering confidence which is calculated by Eq. 4. When $c$ exceeds a predefined threshold $\tau$, the search terminates. We set $\tau = 0.6$ throughout the paper. The implementation of ***RAP*** is shown in Appendix A.

## 5. Experiments

In this section, we evaluate our ***RAP*** on HR benchmarks and a general MLLM benchmark. Further experimental results, including the influence of inference computation scale and the effect of the hyperparameter, are provided in Appendix B. Case studies are illustrated in Figures 8~9 in the Appendix C.

### 5.1. Results on HR Benchmark

**Benchmarks.** We evaluate our ***RAP*** on two HR benchmarks: $V^*$ **Bench** and ***HR-Bench***. $V^*$ **Bench**, derived from SA-1B (Kirillov et al., 2023), averages a resolution of $2246 \times 1582$. More details about ***HR-Bench*** can be found in Sect. 3.1.

**Main Results.** As shown in Table 2, compared to the baseline MLLM, the performance of nearly all models significantly improved with our ***RAP***, demonstrating the model-agnostic trait of ***RAP***. We find that our ***RAP*** can bring significant improvements in both ***FSP*** and ***FCP*** tasks. Our ***RAP*** brings a maximum of 21.0% and 21.7% accuracy improvement on ***HR-Bench 4K*** and ***HR-Bench 8K*** respectively. Additionally, for tasks requiring spatial reasoning capabilities, ***RAP*** demonstrates significant improvements compared to the baseline (*e.g.,* +39.5% accuracy on $V^*$ **Bench** using

*Table 2.* Comparison of our **RAP** (upon several advanced models) with existing works on high-resolution benchmarks. The best performance in each task is in-bold.

| Method | V* Bench | | | HR-Bench 4K | | | HR-Bench 8K | | |
|---|---|---|---|---|---|---|---|---|---|
| | Attribute | Spatial | Overall | FSP | FCP | Overall | FSP | FCP | Overall |
| *Open-source MLLMs* | | | | | | | | | |
| LLaVA-v1.6-7B (Liu et al., 2024b) | 60.9 | 63.2 | 61.8 | 49.0 | 46.8 | 47.9 | 37.3 | 44.3 | 40.8 |
| LLaVA-v1.6-13B (Liu et al., 2024b) | 60.0 | 64.5 | 61.8 | 49.8 | 41.3 | 45.5 | 38.0 | 38.3 | 38.1 |
| LLaVA-v1.6-34B (Liu et al., 2024b) | - | - | - | 55.3 | 50.5 | 52.9 | 44.5 | 50.3 | 47.4 |
| LLaVA-HR-X-13B (Luo et al., 2024) | - | - | - | 61.3 | 46.0 | 53.6 | 49.5 | 44.3 | 46.9 |
| LLaVA-HR-X-7B (Luo et al., 2024) | 51.3 | 64.5 | 56.5 | 57.8 | 46.3 | 52.0 | 42.0 | 41.3 | 41.6 |
| InternVl-1.5-26B (Chen et al., 2024c) | - | - | - | 69.5 | 51.8 | 60.6 | 69.3 | 48.5 | 57.9 |
| Yi-VL-34B (Young et al., 2024) | - | - | - | 46.0 | 42.8 | 44.4 | 39.5 | 38.5 | 39.0 |
| *Closed-source MLLMs* | | | | | | | | | |
| GPT 4o (Hurst et al., 2024) | - | - | 66.0 | 70.0 | 48.0 | 59.0 | 62.0 | 49.0 | 55.5 |
| QWen-VL-max (Bai et al., 2023) | - | - | - | 65.0 | **52.0** | 58.5 | 54.0 | **51.0** | 52.5 |
| *Baseline and RAP* | | | | | | | | | |
| LLaVA-v1.5-7B (Liu et al., 2024a) | 43.5 | 56.6 | 48.7 | 38.5 | 33.8 | 36.1 | 33.0 | 31.3 | 32.1 |
| *-w/ RAP* | **90.4** | **96.1** | **91.1** | 73.8 | 40.5 | 57.1 | 72.3 | 35.3 | 53.8 |
| LLaVA-v1.5-13B (Liu et al., 2024a) | 41.7 | 55.3 | 47.1 | 45.2 | 41.3 | 43.3 | 37.5 | 38.0 | 37.8 |
| *-w/ RAP* | 89.6 | 90.8 | 89.8 | 74.3 | 46.0 | 60.1 | 76.5 | 42.0 | 59.3 |
| LLaVA-ov-0.5B (Li et al., 2024a) | 63.5 | 64.5 | 63.9 | 63.5 | 39.5 | 51.5 | 47.3 | 38.3 | 42.8 |
| *-w/ RAP* | 80.0 | 84.2 | 83.6 | **80.3** | 42.3 | **61.3** | 81.8 | 45.3 | **63.5** |

LLaVA-v1.5-7B). The results show that our method has a clear advantage with HR images.

## 5.2. Results on General Multimodal Benchmark

**Benchmark.** We conduct additional evaluations of **RAP** using the MME-RealWorld (Zhang et al., 2024c), a manually curated benchmark designed for partical, real-world scenarios. This benchmark encompasses five primary categories and 43 sub-class tasks. Due to space constraints, we present results for 9 sub-tasks that exhibit notable performance variations with **RAP**.

**Main Results.** As shown in Table 3, **RAP** improves the performance of LLaVA-v1.5-13B on most sub-tasks, especially on MO/Orientation (+7.3%), AD/Intention (+6.0%), and OCR/license (+10.3%). However, we observe that tasks involving Diagram and Table types do not exhibit significant improvements and, in some cases, even performance degradation. We find that this due to the reliance of such data on the model's spatial awareness and reasoning capabilities, which are inherent limitations of current MLLMs.

## 5.3. Ablation Study

To better understand the role of each module in our **RAP**, we conduct ablation study on **HR-Bench 8K** using LLaVA-v1.5-7B. As shown in Table 4, we first use VisRAG to retrieve key image crops, replacing the original HR images, resulting in an average improvement of 4.5% accuracy compared to

the baseline. However, we find a significant improvement in the **FSP** task, but there is a noticeable performance drop in the **FCP** task. By incorporating the *Spatial-Awareness Layout*, the relative positional relationships between image crops are preserved, leading to an improvement in accuracy on the **FCP** task compared to **+VisRAG**. Finally, we utilize *RE-Search* to determine the optimal $K$ for different samples, resulting in significant improvements in both the **FSP** and **FCP** tasks, with an average improvement of 21.7% accuracy compared to the baseline.

## 5.4. Performance and Efficiency

Efficiency concerns regarding **RAP** may arise among researchers. To address this, Table 5 presents a comparative analysis of throughput and accuracy against SOTA search-based methods (*e.g.,* DC$^2$ and Zoom Eye). **RAP** achieves superior efficiency and performance by directly computing the relevance between image crops and the query, eliminating the need for hierarchical image partitioning, thereby significantly accelerating the search process. More comparison results with search-based methods can be found in Appendix B.5.

## 5.5. Alternative to Model Logit Confidence

In *RE-Search*, we use the logit as a measure of the model's confidence that the current image $V$ can answer the given query. However, for some closed-source models, it is not possible to access the model's output logits. To tackle

*Table 3.* Comparison of the **RAP** against the baseline MLLM on the MME-RealWorld benchmark. MO: Monitoring; AD: Autonomous Driving. The "$\Delta(\uparrow)$" represents the performance gains of our RAP against the baselines.

| Method | MO | | | AD | | | OCR | | |
|---|---|---|---|---|---|---|---|---|---|
| | *Property* | *Orientation* | *Color* | *Intention* | *Attention* | *Visual* | *license* | *Text* | *Address* |
| LLaVA-v1.5-13B | 31.0 | 14.7 | 21.9 | 16.6 | 27.2 | 36.3 | 46.6 | 46.0 | 39.7 |
| *-w/ RAP* | **38.0** | **22.1** | **27.8** | **22.6** | **31.3** | **42.3** | **56.9** | **52.5** | **45.1** |
| $\Delta(\uparrow)$ | +7.0 | +7.3 | +5.9 | +6.0 | +4.2 | +6.0 | +10.3 | +6.5 | +5.4 |

*Table 4.* Ablation study of different module in **RAP**. "**SL**" denotes our *Spatial-Awareness Layout*. We first incorporate VisRAG to retrieve $K$ key image crops, where $K = 8$. Then, we add *Spatial-Awareness Layout* to preserve the relative positional information of the image crops. Finally, we incorporate *RE-Search* to search the optimal $K$.

| | HR-Bench 8K | | | $\Delta(\uparrow)$ |
|---|---|---|---|---|
| | *FSP* | *FCP* | *Overall* | |
| LLaVA-v1.5-7B | 33.0 | 31.3 | 32.1 | - |
| *+ VisRAG* | 52.3 | 25.0 | 38.6 | +6.5 |
| *+ SL* | 50.0 | 27.5 | 38.8 | +6.8 |
| *+ RE-Search* | **72.3** | **35.3** | **53.8** | +21.7 |

*Table 5.* Evaluation of performance and inference efficiency. We analyze the correlation between throughput (samples per minute) and accuracy of LLaVA-v1.5-13B enhanced with our **RAP**, comparing it agains search-based methods on **HR-Bench 4K**.

| Method | Throughput$\uparrow$ | Accuracy$\uparrow$ |
|---|---|---|
| DC$^2$ (Wang et al., 2025) | 2.1 | 51.5 |
| Zoom Eye (Shen et al., 2024) | 3.3 | 58.0 |
| *RAP* | **4.2** | **60.1** |

this issue, we explore a simple alternative approach using generation-based confidence scores. Specifically, we design a scoring prompt that asks the model to evaluate whether the given image contains sufficient information to answer the question. The constructed scoring prompt can be found in Appendix A.4.

We conduct experiments on **HR-Bench** using LLaVA-ov-0.5B. The experimental results are shown in Table 6. Although the generation-based confidence score performs worse than the logit-based confidence score, it still shows a clear improvement over the baseline (achieved an average improvement of 7.4%), demonstrating that generation-based confidence scores through the model can also lead to significant performance gains.

To further investigate the extent to which the generation-based confidence score can replace the logit-based confidence score, we calculate the cosine similarity between the two scores for the same images. As shown in the Figure 4, a high cosine similarity score of 0.97 between the

*Table 6.* Comparison of the performance of different confidence score calculation methods. "**G**" represents the generation-based confidence score and "**L**" represents the logit-based confidence score.

| Method | HR-Bench 4K | HR-Bench 8K |
|---|---|---|
| LLaVA-ov-0.5B | 51.5 | 42.8 |
| w/ *RAP* (G) | 57.0 | 52.1 |
| w/ *RAP* (L) | **61.3** | **63.5** |

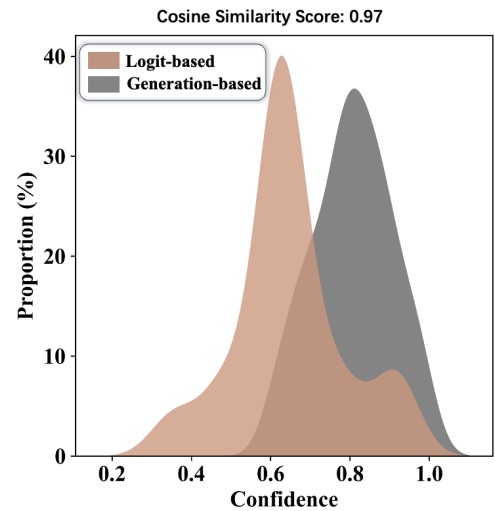

*Figure 4.* Analyzing the distribution of logit-based and generation-based confidence scores.

two types of confidence scores, indicating a remarkable degree of alignment between their distributions. Interestingly, generation-based confidence scores tend to be consistently higher than their logit-based counterparts. Nonetheless, **RAP** utilizing generation-based confidence scores continues to deliver substantial improvements.

### 5.6. Effect of Crop Size

To investigate the impact of crop size, we perform experiments on **HR-Bench 8K** using LLaVA-ov-0.5B. As shown in Table 7, we find that while variations in crop size result in relatively minor differences, all configurations of our **RAP** yield substantial performance gains over the baseline.

*Table 7.* Relationship between crop size and **RAP** performance.

| Method | HR-Bench 8K | | |
| --- | --- | --- | --- |
| | *FSP* | *FCP* | *Avg.* |
| LLaVA-ov-0.5B | 47.3 | 38.3 | 42.8 |
| *w/ RAP* (224 × 224) | **82.0** | 40.8 | 61.4 |
| *w/ RAP* (448 × 448) | 81.8 | **45.3** | **63.5** |
| *w/ RAP* (896 × 896) | 76.8 | 43.3 | 60.0 |

### 5.7. Effect of Retriever

To explore the impact of retrieval quality on **RAP** performance, we conduct experiments on **HR-Bench 8K** using LLaVA-ov-0.5B with SigLIP and VisRAG. Due to the limited text input length of SigLIP, we utilize MLLM to extract noun phrases from the query to compute relevance with image crops.

As shown in Table 8, we evaluate the retrieval quality of SigLIP (Zhai et al., 2023) and VisRAG (Yu et al., 2024), finding that VisRAG achieves superior retrieval performance. Notably, our **RAP** significantly enhances performance even with the relatively weaker retriever (SigLIP); for instance, it delivers a 9.1% overall enhancement on **HR-Bench 8K**.

*Table 8.* Analyzing the relationship between **RAP** performance and retriever using LLaVA-ov-0.5B with SigLIP and VisRAG. The "Params." refer to the total parameters of the retrievers.

| Method | Params. | HR-Bench 8K | | |
| --- | --- | --- | --- | --- |
| | | *FSP* | *FCP* | *Avg.* |
| LLaVA-ov-0.5B | - | 47.3 | 38.3 | 42.8 |
| *w/ RAP (SigLIP)* | 883M | 65.0 | 38.8 | 51.9 |
| *w/ RAP (VisRAG)* | **3.34B** | **81.8** | **45.3** | **63.5** |

### 5.8. Why Does Our Method Work?

Reviewing the design principles of **RAP**: Retrieve image crops related to the query to reduce the image resolution input to the MLLM, thereby enabling the MLLM to perceive images more accurately. To explore the underlying mechanism of **RAP**, we perform experiments that help address the following questions:

*1) Is it truly necessary to retrieve image crops relevant to the query?* we compare randomly retained image crops with query-relevant image crops using LLaVA-v1.5-7B on **HR-Bench 8K**. As shown in Table 9, we randomly retained $K = 4$ and half of the image crops, comparing them with $K$ image crops retrieved through VisRAG that are relevant to the query. The results indicate that *retaining query-relevant image crops is necessary*.

*2) Can RAP accurately select an appropriate $K$?* To an-

*Table 9.* Effect of retrieval on **HR-Bench 8K**. We compare two methods: randomly retaining $K$ image crops (**Random**) and retrieving $K$ image crops. The "half" refers to retaining half of the image crops (**Retrieval**).

| Method | HR-Bench 8K | | |
| --- | --- | --- | --- |
| | *FSP* | *FCP* | *Avg.* |
| *Random* ($K = 4$) | 25.0 | 23.8 | 24.4 |
| *Random* ($K = half$) | 29.0 | 24.5 | 26.8 |
| *Retrieval* ($K = 4$) | **52.3** | **25.0** | **38.6** |

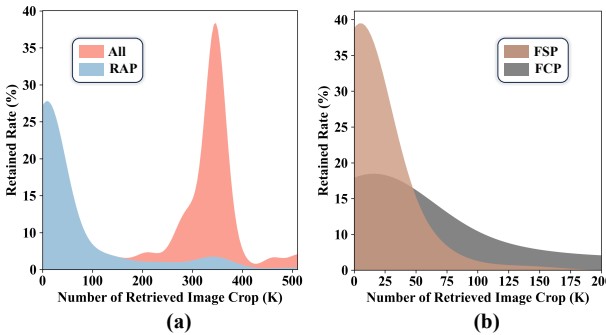

*Figure 5.* Analyzing the distribution for selecting $K$ using our **RAP**. (a) The distribution of $K$ selected by **RAP**, where "All" denotes the total number of image crops in the original image. (b) The distribution of $K$ corresponding to different task types.

swer this question, we visualize the distribution of the number of retrieved image crops ($K$) for LLaVA-v1.5-7B w/ **RAP** on **HR-Bench 8K**. As shown in Figure 5(a), our **RAP** effectively reduces the number of image crops, resulting a +21.7% accuracy improvement. Additionally, for the **FSP** task, the $K$ selected by our **RAP** is smaller, while for the **FCP** task, it is widely distributed across the range corresponding to larger $K$ (*e.g.,* $K \geq 60$). The experiment results demonstrate that *our RAP can provide accurate $K$, thereby effectively reducing the image resolution.*

## 6. Conclusion

In this paper, we propose a novel training-free framework **Retrieval-Augmented Perception (RAP)** to enhance HR image understanding in MLLMs. We empirically demonstrated the effectiveness and universality of **RAP** on several widely used MLLM benchmarks. From the results, we mainly conclude that: (1) Retrieving image crops relevant to the query can result in significant improvements; (2) Maintaining the relative spatial relationships of the retrieved image crops is essential, particularly for tasks that rely on positional information; (3) The number of image crops that need to be retained varies across different task types. In our future work, we will explore more token compression techniques to further enhance HR perception and efficiency.

## Impact Statement

Our work introduces a training-free framework, *Retrieval-Augmented Perception (RAP)*, which significantly improves the ability of multimodal large language models (MLLMs) to perceive and reason over high-resolution (HR) images. Specifically, by combining retrieval techniques with efficient spatial-awareness and adaptive search, **RAP** enables MLLMs to capture finer visual details and spatial relationships in images of resolutions previously unattainable with standard approaches.

## Acknowledgements

This project is supported by the National Key Research and Development Program of China (2023YFC2705700), the National Natural Science Foundation of China (Grant No. U23A20318, 62276195 and 62225113), the Science and Technology Major Project of Hubei Province (Grant No. 2024BAB046), the Foundation for Innovative Research Groups of Hubei Province (Grant No. 2024AFA017). This project is supported by the National Research Foundation, Singapore, under its NRF Professorship Award No. NRF-P2024-001.

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

# Appendix

This appendix presents a detailed description of the proposed **Retrieval-Augmented Perception (RAP)**, along with additional results from **comprehensive experiments**, **ablation studies**, and **case analyses**. The structure of the appendix is summarized as follows.

➤ Appendix A provides the implementation details of **RAP**, including preprocessing and hyperparameter settings. Specifically, we introduce the specific implementation of the *Spatial-Awareness Layout* in Appendix A.1. Appendix A.2 presents the details of *RE-Search*, including the number of search steps and termination conditions. Appendix A.3 outlines the algorithmic workflow of the proposed **RAP**. Finally, the prompt used for generation-based confidence calculation is provided in Appendix A.4.

➤ Appendix B provides additional experimental results, including the experiment results on widely used benchmarks (Appendix B.1), with more powerful MLLMs (Appendix B.2), the impact of inference scaling (Appendix B.3), the influence of hyperparameters (Appendix B.4), and a comprehensive comparison with search-based methods (Appendix B.5).

➤ Appendix C provides a qualitative analysis of the proposed **RAP** and the current SOTA methods (Wang et al., 2025; Shen et al., 2024). Appendix C.1 presents qualitative analysis examples on the *fine-grained single-instance perception task*, while Appendix C.2 illustrates examples on the *fine-grained cross-instance perception task*.

➤ Appendix D presents and discusses the limitations of our **RAP**, providing directions for future research.

## A. Additional Implement Details

Due to space constraints in the main paper, additional implementation details are provided in this section. In Appendix A.1 and Appendix A.2, we elaborate on the implementation of *Spatial-Awareness Layout* and *Retrieved-Exploration Search*, respectively. Building upon these components, Appendix A.3 presents the complete algorithmic workflow of **Retrieval-Augmented Perception (RAP)**.

### A.1. Implement Details of Spatial-Awareness Layout

Given an input image $I$, it is first partitioned into smaller image crops based on a predefined crop size, which corresponds to the preferred resolution of the retriver. To ensure that the image size are divisible by the crop size and to prevent potential loss of visual information, padding is applied to the original image $I$ as necessary. Next, only non-zero image crops (referred to as *valid image crops*) are retained to eliminate potential interference in the subsequent semantic similarity computation with the query. The retriever, specifically VisRAG (Yu et al., 2024) in this implementation, is then utilized to compute the cosine similarity between the user-provided query and each valid image crop. Based on the given $K$, the top $K$ image crops with the highest similarity scores are selected. Subsequently, the rows and columns containing the selected crops are retained, forming a compressed matrix $M'$. Finally, using $M'$, the corresponding image crops in $I$ are mapped to construct a new transformed image $I'$.

### A.2. Implement Details of Retrieved-Exploration Search

In *Retrieved-Exploration Search (RE-Search)*, a given HR image $I$ is designated as the root node for the search process. The semantic similarity between $I$ and the query is computed, along with an assessment of whether the image contains sufficient information for the MLLM to generate an appropriate response to the query. Subsequently, different proportions of image crops are retained based on the predefined retention ratio set $P$. Specifically, for each node, $25\%, 50\%, and 75\%$ of the image crops are preserved. The REward function is then applied to the retained image crops, and corresponding child nodes are created. These child nodes are added to the list of candidate nodes $\mathcal{O}$ for further exploration. Throughout the search process, the algorithm continuously tracks and maintains the optimal node identified thus far. The search process terminates if the current search step exceeds the predefined maximum search steps, which is set to 200 by default, or when the answering confidence $c$ of the current node surpasses a specified threshold $\tau$. We set $\tau = 0.6$ throughout the paper.

### A.3. Complete of Algorithm Workflow

With the above notations and definitions in place, we provide the complete algorithm workflow in Algorithm 2.

**Algorithm 2** *Retrieval-Augmented Perception*

---

**Require:** HR image $I$, Retriever $R$, Retention ratio $P$, Max steps $max_s$

  import $SpatialLayout$ from Algorithm 1

  **function** REward($q, I, d$)

    $V : \{v_1, ..., v_n\} \leftarrow$ Divide image $I$ into image crops

    $g \leftarrow s(q, V)$

    $h \leftarrow 1 - \mathcal{P}_\theta(p_h(q), I)$

    $w \leftarrow (1 - b) \cdot (1 - \frac{1}{d})^2 + b$

    $f \leftarrow (1 - w) \cdot g + w \cdot h$

    return $f$

  **end function**

  **function** $RetrievalSubNode(V)$

    $V : \{v_1, ..., v_n\} \leftarrow$ Divide image $I$ into image crops

    Initialize $V_s \leftarrow \emptyset$

    Initialize $M_s \leftarrow \emptyset$

    **for** $idx = 1$ to $|P|$ **do**

      $S \leftarrow s(q, V)$

      $V', M \leftarrow topK(S, V, P[idx])$

      $V_s \leftarrow V_s \cup \{V'\}$

      $M_s \leftarrow M_s \cup \{M\}$

    **end for**

    return $V_s, M_s$

  **end function**

  $f \leftarrow REward(q, I, 0)$

  $t_0 \leftarrow$ Node($v = I, f = f, d = 0$)

  Initialize $\mathcal{O} \leftarrow \{t_0\}$

  $t_{optimal} \leftarrow t_0$

  $S \leftarrow 0$  /*Current step */

  **while** $\mathcal{O}$ is not empty and $S \leq max_s$ **do**

    Extract all confidence $F \leftarrow [\, o.f : o \in \mathcal{O} \,]$

    $idx \leftarrow \arg\min(F)$

    $t_s \leftarrow \mathcal{O}[idx]$

    Remove $t_s$ from $\mathcal{O}$

    $S \leftarrow S + 1$

    **if** $t_s.f > \tau$ **then**

      return $t_{optimal}.v$

    **end if**

    /* Retrieval*/

    $V_s, M_s \leftarrow RetrievalSubNode(I)$

    /* Exploration*/

    **for** $idx = 1$ to $|V_s|$ **do**

      $I_s \leftarrow SpatialLayout(V_s[idx], M_s[idx])$

      $f_s \leftarrow REward(q, I_s, t_s.d)$

      $t_{s+1} \leftarrow$ Node($v = I_s, f = f_s, d = t_s.d + 1$)

      $\mathcal{O} \leftarrow \mathcal{O} \cup \{t_{s+1}\}$

      **if** $t_{s+1}.f > t_{optimal}.f$ **then**

        $t_{optimal} \leftarrow t_{s+1}$

      **end if**

    **end for**

  **end while**

---

## A.4. Prompt for Generation-based Confidence Scores

To improve the applicability of our **RAP** method to closed-source MLLMs, we investigate a straightforward alternative that leverages generation-based confidence scores. Specifically, we employ the prompt shown in Table 10 to prompt the MLLM to generate a confidence score conditioned on the given input image and question.

*Table 10.* Prompt template for generation-based confidence scores. The "**[question]**" is placeholder meant to be replaced with specific question from the dataset.

| **Prompt Template for Generation-based Confidence Score** |
| --- |
| Question: **[question]** |
| Could you answer the question based on the available visual information? Return only a JSON object with a numerical confidence score (0 10) of "Yes" like {"Yes": x}. |

# B. More Experiment Result

## B.1. More Experiment Results on Widely used Benchmarks

We conduct additional experiments on five widely used benchmarks: DocVQA (Mathew et al., 2021), ChartQA (Masry et al., 2022), TextVQA (Singh et al., 2019), AI2D (Hiippala et al., 2021), and MMStar (Chen et al., 2024b). As shown in Table 11, incorporating **RAP** led to performance of 1.8% and 2.1% on LLaVA-v1.5 7B and 13B, respectively. We also observed taht **RAP** brings more notable improvements on higher-resolution images. For example, on DocVQA, which has an average resolution of $1599 \times 1241$, **RAP** improved performance by 4.7% and 2.3% for LLaVA-v1.5 7B and 13B, respectively.

*Table 11.* Comparison of the **RAP** against the baseline MLLM on five widely used benchmarks.

|  | **DocVQA** | **ChartQA** | **TextVQA** | **AI2D** | **MMStar** |
| --- | --- | --- | --- | --- | --- |
| LLaVA-v1.5-7B | 21.5 | 18.2 | 45.8 | 54.9 | 30.3 |
| w/ **RAP** | **26.2** | **18.5** | **46.8** | **55.1** | **33.1** |
| LLaVA-v1.5-7B | 23.7 | 18.5 | 49.0 | 60.2 | 32.8 |
| w/ **RAP** | **26.0** | **23.2** | **50.0** | **60.9** | **34.4** |

## B.2. More Experiment Results with Powerful MLLMs

To further demonstrate the effectiveness and generalizability of our **RAP**, we conduct experiments on HR-Bench using several advanced MLLMs, including Oryx-1.5-7B, CogVLM-LLama3-19B, Cambrian-8B and LLaVA-ov-72B. As shown in Table 12, our **RAP** consistently boosts performance across all models. These results underscore the robustness of our **RAP** across a wide range of model architectures, highlighting its potential as a universal enhancement for high-resolution image perception.

*Table 12.* Comparison of our RAP with Advanced MLLMs on HR-Bench.

|  | **HR-Bench 4K** | | | **HR-Bench 8K** | | |
| --- | --- | --- | --- | --- | --- | --- |
|  | *FSP* | *FCP* | *Avg.* | *FSP* | *FCP* | *Avg.* |
| Oryx-1.5-7B (Liu et al., 2024d) | 62.0 | 50.5 | 56.3 | 53.8 | **45.3** | 49.5 |
| *w/ RAP* | **80.8** | **52.5** | **66.6** | **77.3** | 45.2 | **61.3** |
| CogVLM-LLama3-19B (Hong et al., 2024) | 69.5 | **48.8** | 59.1 | 53.3 | **45.0** | 49.1 |
| *w/ RAP* | **82.5** | 48.3 | **65.4** | **76.3** | 43.7 | **60.0** |
| Cambrian-8B (Tong et al., 2024) | 45.5 | **45.5** | 45.5 | 30.8 | **45.0** | 37.9 |
| *w/ RAP* | **70.0** | 44.3 | **57.1** | **66.5** | 44.8 | **55.6** |
| LLaVA-ov-72B (Li et al., 2024a) | 76.8 | 57.5 | 67.1 | 70.5 | **55.3** | 62.9 |
| *w/ RAP* | **91.5** | **60.0** | **75.8** | **89.3** | 54.5 | **71.9** |

## B.3. RAP Performance and Inference Computation Scale

To analyze the performance changes with different search steps, we plot the performance of RE-Search steps. We conduct experiments on **HR-Bench 8K** using LLaVA-v1.5 7B & 13B, LLaVA-ov-0.5B. To accurately analyze the relationship between search steps and performance, we set $\tau = \infty$ to prevent early termination due to threshold constraints during the search process. This forces the model to perform a fixed number of steps and selects the $K$ with the lowest cost as the final output. As shown in Figure 6, we observe that increasing the number of search steps improves the performance,

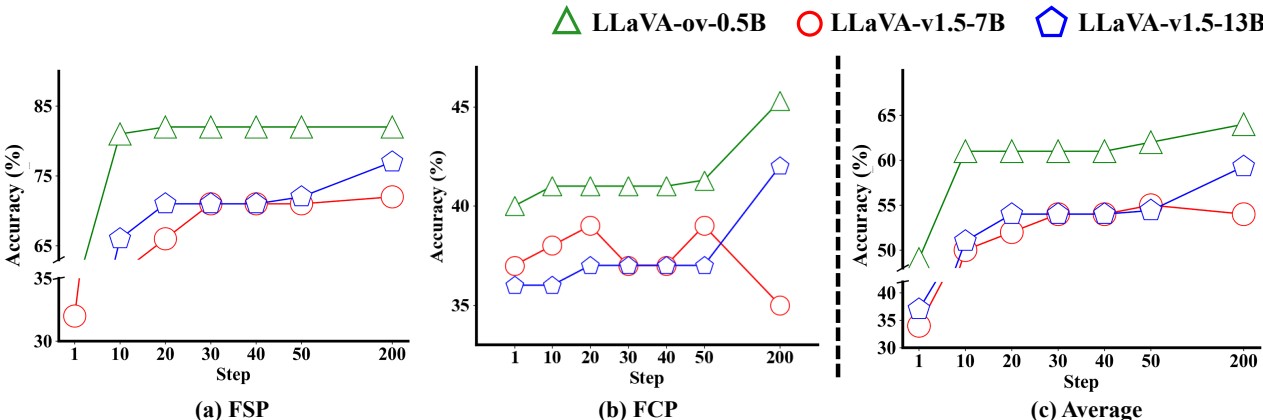

Figure 6. Performance vs. *RE-Search* steps on **HR-Bench 8K. (a) Fine-grained Single-instance Perception (FSP); (b) Fine-grained Cross-instance Perception (FCP); (c) Overall Performance.**

especially on the **FCP** task. However, the gains are marginal for LLaVA-v1.5-7B but more pronounced for stronger models like LLaVA-ov-0.5B and LLaVA-v1.5-13B. Our analysis reveals that the **FCP** task requires consideration of the spatial relationships between image crops and their spatial combinations, making capabilities result in a more noticeable performance improvement with increased search steps.

## B.4. Effect of bias $b$

In the *RE-Search*, we use $w$ to balance the cost $g(\cdot)$ and heuristic function $h(\cdot)$ in different depth. In Eq. 6, we use $b$ as the bias value to control the influence of depth on $w$. A smaller $b$ indicates a greater influence of depth on $w$, while a larger $b$ reduces this influence.

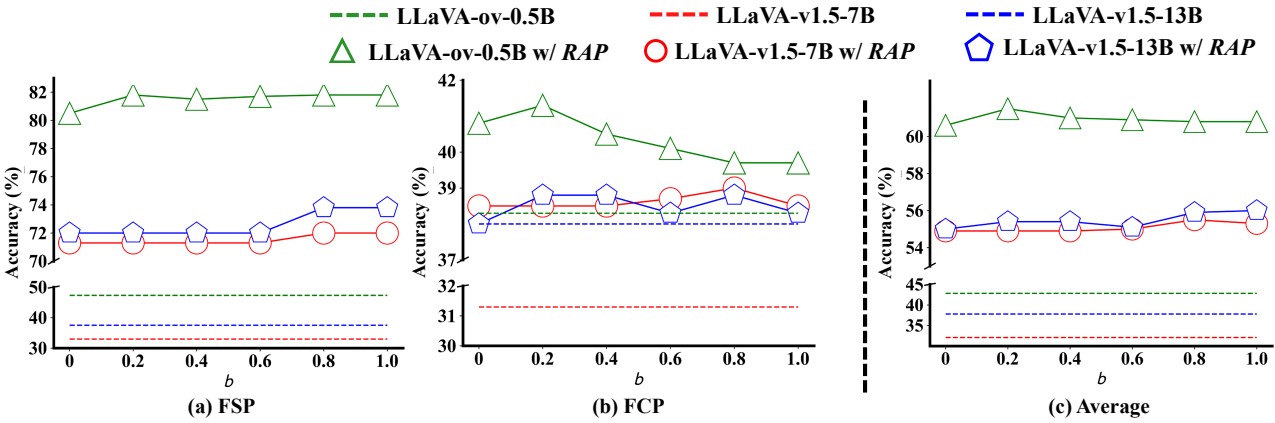

Figure 7. Impact of bias value $b$, illustrating how the accuracy changes when varying bias value $b$.

As shown in Figure 7, our **RAP** is not sensitive to the value of $b$, and it consistently outperforms the baseline across all

configurations. Furthermore, we observe that smaller values of $b$ lead to better results for all models. Therefore, to ensure a fair comparison, we set $b = 0.2$ by default.

### B.5. Compared with Other HR Processing Methods

We compare our **RAP** with two HR processing methods – $DC^2$ and Zoom Eye. $DC^2$ is a training-free framework to enhance MLLM understanding of HR images by partitioning images, generating textual descriptions for image crops, and integrating them for improved perception. Zoom Eye, a tree search algorithm, is designed to effectively navigate the hierarchical and visual structures of images to extract relevant information.

As shown in Table 13, compared with the baseline, all HR processing methods bring the average performance gains. Among all these methods, our **RAP** achieves the relatively better formance on most tasks. For instance, **RAP** achieve an accuracy of 73.8% and 72.3% on **HR-Bench 4K** and **HR-Bench 8K**, respectively, using LLaVA-v1.5-7B, representing improvements of 6.0% and 6.8% compared to Zoom Eye. These results can prove the superiority of our **RAP**.

*Table 13.* Performance comparison between **RAP** and other HR methods. We conduct experiments on $V^*$ **Bench** and **HR-Bench** using LLaVA-v1.5 7B and 13B. The "$\Delta(\uparrow)$" represents the performance gains of HR methods against the baselines.

| Method | $V^*$ Bench | | | HR-Bench 4K | | | HR-Bench 8K | | | $\Delta(\uparrow)$ |
|---|---|---|---|---|---|---|---|---|---|---|
| | Attribute | Spatial | Overall | FSP | FCP | Overall | FSP | FCP | Overall | |
| LLaVA-v1.5-7B | 43.5 | 56.6 | 48.7 | 38.5 | 33.8 | 36.1 | 33.0 | 31.3 | 32.1 | - |
| *-w/ $DC^2$* | 49.6 | 59.2 | 51.6 | 45.3 | 37.0 | 41.1 | 36.5 | 33.3 | 34.9 | +2.5 |
| *-w/ Zoom Eye* | 83.5 | 82.9 | 83.3 | 67.8 | 38.8 | 53.3 | 65.5 | **36.0** | 50.8 | +22.5 |
| *-w/ RAP* | **90.4** | **96.1** | **91.1** | **73.8** | **40.5** | **57.1** | **72.3** | 35.3 | **53.8** | +27.0 |
| LLaVA-v1.5-13B | 41.7 | 55.3 | 47.1 | 45.2 | 41.3 | 43.3 | 37.5 | 38.0 | 37.8 | - |
| *-w/ $DC^2$* | 54.8 | 56.6 | 57.3 | 52.0 | **51.0** | 51.5 | 40.0 | 41.0 | 40.5 | +7.1 |
| *-w/ Zoom Eye* | 87.5 | 81.6 | 85.3 | 73.0 | 43.0 | 58.0 | 67.3 | **45.5** | 56.4 | +23.9 |
| *-w/ RAP* | **89.6** | **90.8** | **89.8** | **74.3** | 46.0 | **60.1** | **76.5** | 42.0 | **59.3** | +27.0 |

## C. Case Study

### C.1. Qualitative Examples of Fine-grained Single-instance Perception Task

Figure 8 illustrates two instances where incorporating different HR processing methods ($DC^2$, Zoom Eye and our **RAP**) on LLaVA-v1.5-13B. In the first example, the critical information "08-26" in the image lies exactly at the boundary of two image crops. Zoom Eye retains only a part of it, leading to the loss of critical information. In contrast, our **RAP**, leaveraging *RE-Search*, accrately preserves the critical information and provides a correct response. In the second example, $DC^2$ initially searches along an incorrect path, resulting in an erroneous final answer. In contrast, our **RAP** method accurately retrieves the cup on the ground, thereby providing the correct answer.

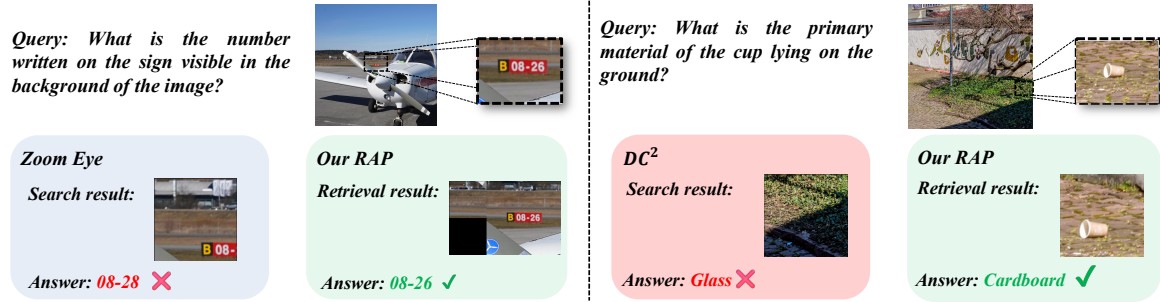

*Figure 8.* Qualitative examples of **Fine-grained Single-instance Perception** task. We conduct experiments on **HR-Bench 4K** using LLaVA-v1.5-13B with HR processing methods.

## C.2. Qualitative Examples of Fine-grained Cross-instance Perception Task

Figure 9 presents two examples demonstrating the performance of different HR processing methods ($DC^2$, Zoom Eye, and our **RAP**) applied to LLaVA-v1.5-13B. In the first example, Zoom Eye fails to consider the spatial relationships between image crops, leading to an incorrect search result and an erroneous response. In contrast, our **RAP** effectively preserves the relative positions between image crops, enabling the generation of a correct answer. In the second example, multiple image crops are required for accurate reasoning. However, $DC^2$ retrieves only a single image crop based on the keyword "chair" from the query, resulting in an incorrect answer. In contrast, our **RAP** accurately retains the critical image crops, thereby producing the correct answer.

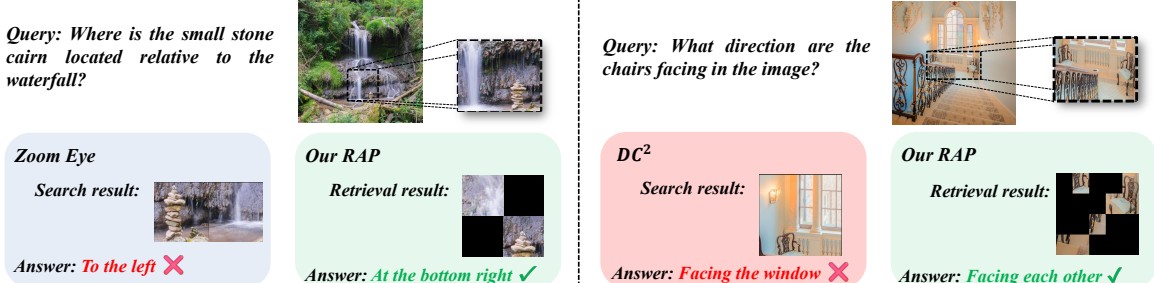

*Figure 9.* Qualitative examples of ***Fine-grained Cross-instance Perception*** task. We conduct experiments on ***HR-Bench 4K*** using LLaVA-v1.5 13B with HR processing methods.

# D. Limitation Discussion

Admittedly, the proposed **RAP** has limitations, despite its promising performance on HR benchmarks. In particular, the proposed method relies on an external retriever for plausible results. In our future work, we will strive to address this limitation by investigating the model's internal visual perception to adaptively select key image crops.

