# OpenReview forum: "Retrieval-Augmented Perception: High-resolution Image Perception Meets Visual RAG"
_ICML.cc/2025/Conference — ICML 2025 oral_

### Official Review · Reviewer_TfPd · 2025-03-03

**Overall Recommendation:** 4

**Summary:**

- The paper introduces a RAG-based approach to handle High-Resolution visual reasoning with MLLMs.
- The authors develop a Retrieval-Augmented Perception (RAP) framework to find the important image crops required for answering a given query based on the query-crop similarity and using those for inference.
RAP is an iterative process with two stages: (I) RE-Search (to find the optimal number of crops using confidence scores about the cropped image to answer the query from multiple MLLM feed-forward passes) and (ii) Spatial-Awareness Layout (to lay out the most similar crops while maintaining their relative positioning from the original image).
- The authors present an extensive analysis of factors affecting performance, including the number of retrieved image crops and their layout.
- The authors test their framework on the MME-Realworld, V*, and HR-Bench benchmarks and show improvements with different base MLLMs from the LLaVA series.

## update after rebuttal

I am raising my score to accept based on the rebuttal.

**Claims And Evidence:**

- The authors claim RAP improves performance on high-resolution benchmarks over standard MLLMs built with fixed-resolution visual encoders, which are supported by experiments.
- The authors claim RAP is more efficient than other search-based retrieval methods, which is also well supported.

**Essential References Not Discussed:**

Nope

**Experimental Designs Or Analyses:**

- The authors analyze the different components of their framework:
   - Their SL and RE-Search techniques bring on improvements along with the VisRAG method.
- The authors use the LLaVA-1.5-7B/13B and LLaVA-OV-0.5B models to test their method and show improvements. However, the mentioned models are not the best, and the community rarely uses these for their applications. To show the practical effectiveness of the approach, the authors should consider testing it with more powerful MLLMs like LLaVA-OV-7B/72B and Cambrian-8B/34B.
 - Related to the above point, Cambrian is a multi-encoder framework that uses spatial compression on CLIP-ConvNeXT features, which are better for high-res images, so this is an important experiment, IMO.

**Methods And Evaluation Criteria:**

- The authors evaluate on high res benchmarks which is aligned with their RAG-based framework design motivation

**Other Comments Or Suggestions:**

N/A

**Other Strengths And Weaknesses:**

- Did the authors test their framework on other widely benchmarks like DocVQA, ChartQA, TextVQA, AI2D, or any ocr/document-related tasks?
- The authors should compare their results to a native high-res MLLM like Oryx-MLLM (https://github.com/Oryx-mllm/Oryx).
- The authors show CogVLM has high performance in Fig. 1, but they don't include it in Tab.2 or share results with RAP with CogVLM as the base MLLM. why? I believe a more thorough evaluation with different benchmarks/models is important to show the complete effectiveness of the method.
- The authors do not report the throughput for the original LLaVA-1.5 in Tab. 5 and add other better general MLLMs like LLaVA-ov-72B/CogVLM/Oryx-MLLM to the mix for a better understanding of RAP's practical usefulness.

**Questions For Authors:**

N/A

**Relation To Broader Scientific Literature:**

- High-resolution visual reasoning is an important problem and highly relevant to the current literature as also reflected in the design of benchmarks like V* and HR-Bench.

**Theoretical Claims:**

- There are no theoretical claims made in the paper, IMO.

---

> ### Author Rebuttal · Authors · 2025-04-01
>
> We would like to thank the reviewer for the constructive comments and suggestions.
>
> > **Q1:** Experiments on widely used benchmarks.
>
> **R:** We appreciate the reviewer's suggestion. As suggested, we conduct additional experiments on five widely used benchmarks: **DocVQA, ChartQA, TextVQA, AI2D, and MMStar**. As shown in the table below, incorporating *RAP* led to performance of 1.8% and 2.1% on LLaVA-v1.5 7B and 13B, respectively. We also observed that *RAP* brings more notable improvements on higher-resolution images. For example, on DocVQA, which has an average resolution of $1599\times 1241$, *RAP* improved performance by 4.7% and 2.3% for LLaVA-v1.5 7B and 13B, respectively.
>
> |  | DocVQA   | ChartQA  | TextVQA  | AI2D | MMStar |
> | -- | -- | -- | -- | -- | -- |
> | LLaVA-v1.5-7B  | 21.5     | 18.2     | 45.8 | 54.9     | 30.3     |
> | **LLaVA-v1.5-7B w/ *RAP***     | **26.2** | **18.5** | **46.8** | **55.1** | **33.1** |
> | LLaVA-v1.5-13B | 23.7 | 18.5 | 49.0 | 60.2 | 32.8  |
> | **LLaVA-v1.5-13B w/ *RAP***     | **26.0** | **23.2** | **50.0** | **60.9** | **34.4** |
>
> > **Q2:** Experiments with more powerful MLLMs.
>
> **R:** We thank the reviewer for the constructive advice. To further demonstrate the effectiveness and generalizability of our *RAP*, we conduct experiments on HR-Bench using several advanced MLLMs, including **Oryx-1.5-7B, CogVLM-LLama3-19B, Cambrian-8B, LLaVA-ov-7B and 72B**. As shown in the table below, our *RAP* consistently boosts performance across all models. These results underscore the robustness of our *RAP* across a wide range of model architectures, highlighting its potential as a universal enhancement for high-resolution image perception. We will include the corresponding results and discussions in the revised version.
>
> | | HR-Bench 4K | HR-Bench 8K |
> | -- | -- | -- |
> | Oryx-1.5-7B | 56.3 | 49.5  |
> | **Oryx-1.5-7B w/ *RAP***   | **66.6**  | **61.3** |
> | CogVLM-LLama3-19B | 59.1  | 49.1 |
> | **CogVLM-LLama3-19B w/ *RAP*** | **65.4**    | **60.0** |
> | Cambrian-8B | 45.5 | 37.9 |
> | **Cambrian-8B w/ *RAP***  | **57.1**    | **55.6** |
> | LLaVA-ov-7B  | 63.0 | 49.3 |
> | **LLaVA-ov-7B w/ *RAP*** | **70.3** | **63.8** |
> | LLaVA-ov-72B | 67.1  | 62.9 |
> | **LLaVA-ov-72B w/ *RAP***  | **75.8**  | **71.9** |
>
> > **Q3:** Comparison with native high-res MLLMs.
>
> **R:** We appreciate the reviewer for the comments. We'd like to clarify that due to limitations in visual encoders and LLMs' long-context handling, current MLLMs struggle with high-resolution (HR) image perception. Existing native high-res MLLMs mainly address this through two strategies:
>
> - **Splitting HR images into multiple image crops**
>
>   These methods split HR images into smaller crops, encode them separately, and concatenate the features for the LLM. Oryx uses OryxViT to handle arbitrary resolutions, but processing 8K images produces up to 300K visual tokens, **heavily taxing the LLM's long-context capacity**. In practice, this often led to erroneous outputs, so we downsampled to 2K resolution (as recommended), which unfortunately resulted in loss of important visual details.
>
> - **Using HR visual encoder**
>
>   Another line of work explores HR visual encoders designed to handle HR images. For instance, ConvNeXt-L supports inputs up to $768\times 768$ resolution. However, such encoders still **downsample 8K images to align with pretraining resolutions, limiting their effectiveness on HR images**. Even Cambrian, a multi-encoder framework, only supports resolutions up to $384\times 384$, necessitating downsampling for 8K inputs and resulting in significant loss of critical visual information.
>
> Inspired by RAG's success in improving long-context capabilities in LLMs, we adapt it to MLLMs for HR image perception. *RAP* retrieves and preserves only the most relevant image crops, reducing input resolution while retaining essential information. Our *RAP* can be integrated into any MLLMs, significantly enhancing their perception of HR images. For example, LLaVA-v1.5 7B with *RAP* reaches 53.8% accuracy on HR-Bench 8K, outperforming Oryx-1.5-7B (49.5%) and Cambrian (37.9%), validating *RAP*'s effectiveness.
>
> > **Q4:** The practical usefulness of our *RAP*.
>
> **R:** We'd like to thank the reviewer for the comments. Here, to robustly demonstrate the practical usefulness of our *RAP*, we evaluate its impact on both throughput and accuracy. As shown in the table below, LLaVA-ov-0.5B equipped with *RAP* achieves performance comparable to the much larger LLaVA-ov-72B, yet operates with **2.3 times higher throughput and requires 22 times fewer parameters**. These results highlight *RAP*'s potential for efficient, high-performance deployment, particularly in resource-constrained environments such as edge devices.
>
> |  | Params | Throughput | Accuracy |
> | -- | - | -- | -- |
> | LLaVA-ov-0.5B   | 0.5B   | **40.0**   | 42.8     |
> |  **LLaVA-ov-0.5B w/ *RAP (VisRAG)*** | 3.3B   | 20.0  | **63.5** |
> | LLaVA-ov-72B    | 72B    | 8.7  | 62.9 |

---

### Official Review · Reviewer_qqJL · 2025-03-10

**Overall Recommendation:** 4

**Summary:**

This paper proposes Retrieval-Augmented Perception (RAP), a training-free framework that enhances high-resolution image perception in multimodal large language models by leveraging RAG. RAP retrieves and fuses relevant image crops while maintaining their spatial relationships using a Spatial-Awareness Layout and dynamically determines the optimal number of retrieved crops through Retrieved-Exploration Search (RE-Search). Experiments show that RAP brings consistent improvement on HR benchmarks and general MLLM tasks.

## update after rebuttal
Thanks for answering my questions. I will keep my score.

**Claims And Evidence:**

The claims are supported by clear and convincing evidence according to experimental results.

**Essential References Not Discussed:**

Although it can be considered as concurrent work, the authors are encouraged to include the latest papers [a, b] from ICLR2025 in the related work section.
[a] SV-RAG: LoRA-Contextualizing Adaptation of MLLMs for Long Document Understanding
[b] MMed-RAG: Versatile Multimodal RAG System for Medical Vision Language Models

**Experimental Designs Or Analyses:**

The experiments are conducted on high-resolution benchmarks and general benchmarks, with LLaVA-v1.5 7B & 13B and LLaVA-ov-0.5B as baselines for comparison. The metrics include accuracy and throughput. The experiments confirm the validity and effectiveness of the proposed method.

**Methods And Evaluation Criteria:**

The proposed method enhances the perception of high-resolution images by MLLMs through retrieval augmentation, without requiring additional training. As a plug-in, it can theoretically be applied to any MLLMs. This is meaningful for the current MLLM's perception of high-resolution images. However, it may incur additional inference overhead.

**Other Comments Or Suggestions:**

Please consider to add citations to the papers mentioned above.

**Other Strengths And Weaknesses:**

Strengths:
+ The paper explores applying RAG to MLLMs for the perception of high-resolution images. Through comprehensive experiments, it investigates the impact of the layout of retrieved image crops and the number of retrieved image crops on the final performance.
+ This paper contributes to the ability of MLLMs to perceive high-resolution images.
+ Experimental results show the effectiveness of the proposed methods.

Weaknesses:
- The method requires the introduction of an additional retriever, which consumes certain computational resources.
- Since the method relies on the model's confidence score regarding whether currently retrieved image crops can answer the question, it cannot be directly applied to closed-source models.

**Questions For Authors:**

Although the authors emphasize that RAP is a RAG-based method, it also uses search techniques to enhance MLLMs' perception of high-resolution images. What is the fundamental difference between RAP and search-based methods?

**Relation To Broader Scientific Literature:**

The method seems to be similar to RAG applied to LLMs [a], enhancing the ability of LLMs to perceive long-context in NLP.

[a] Long-Context LLMs Meet RAG: Overcoming Challenges for Long Inputs in RAG

**Theoretical Claims:**

The paper does not present particularly complex techniques, with its effectiveness mainly demonstrated through experimental results. The ablation study in Table 4 and Figure 4 effectively emphasizes the validity of the method's design.

---

> ### Author Rebuttal · Authors · 2025-04-01
>
> We truly appreciate the reviewer for the thoughtful comments and suggestions, as well as the positive support.
>
> > **Q1:** The concern about the extra retriever increasing computation.
>
> **R:** We appreciate the reviewer's comments. Although our *RAP* includes an additional retriever module, it still delivers a significant performance gain despite the overhead. As shown in the table below, our evaluation on HR-Bench 8K demonstrates that integrating *RAP* with LLaVA-ov-0.5B, using VisRAG as the retriever, results in a total model size only 3.3 billion parameters. Despite this lightweight configuration, it achieves an impressive 20.7% improvement in accuracy. Notably, **LLaVA-ov-0.5B w/ *RAP* matches the performance of LLaVA-ov-72B while using approximately 22 times fewer parameters**, highlighting *RAP*'s efficiency, scalability, and strong potential for high-resolution visual understanding.
>
> |                 | Params | Throughput | Accuracy |
> | --------------- | ------ | ---------- | -------- |
> | LLaVA-ov-0.5B   | 0.5B   | **40.0**   | 42.8     |
> | LLaVA-ov-0.5B w/ ***RAP (VisRAG)*** | 3.3B   | 20.0       | **63.5** |
> | LLaVA-ov-72B    | 72B    | 8.7        | 62.9     |
>
>
>
> > **Q2:** Replace confidence score calculation in *RE-Search*.
>
> **R:** We appreciate the reviewer's point. To clarify, the OpenAI API [R1] currently offers a `logprobs` parameter, which returns the log probabilities of output tokens — an essential feature we leverage in our *RE-Search* module to compute confidence scores.
>
> Beyond this, we also explore an alternative approach using **generation-based confidence scoring**. Specifically, we prompt the MLLM to self-assess whether the image provides sufficient information to answer the question. The constructed prompt is as follows:
>
> ```Question: {}\nCould you answer the question based on the available visual information? Return only a JSON object with a numerical confidence score (0~10) of "Yes"  like {"Yes": x}.```
>
> We normalize the final score to the range $ [0,1] $ for consistency. As shown in the table below, incorporating this generation-based score into *RAP* resulted in a **7.4%** performance gain over the baseline, demonstrating its effectiveness.
>
> The relevant results and discussions will be incorporated into the revised version.
>
> |            | HR-Bench 4K | HR-Bench 8K |
> | ---------------------- | ----------- | ----------- |
> | LLaVA-ov-0.5B                              | 51.5        | 42.8        |
> | **LLaVA-ov-0.5B w/ *RAP* (logit-based confidence score)**  | **61.3**    | **63.5**    |
> | LLaVA-ov-0.5B w/ *RAP* (generation-based confidence score) | 57.0        | 52.1        |
>
>
> [R1] OpenAI API, https://platform.openai.com/docs/api-reference/completions/create
>
>
>
> > **Q3:** What is the fundamental difference between *RAP* and search-based methods?
>
> **R:** We thank the reviewer for the constructive comments. We would like to clarify the main differences between our *RAP* and search-based methods as follows:
>
> - **Search-based methods yield high-resolution inputs with multiple crops, while RAP lowers resolution by keeping only key spatially-aware crops**
>
>   Search-based methods model high-resolution images using a tree structure and apply search algorithms to identify key image crops. However, when a question requires information from multiple crops, methods like $DC^2$ tend to select the LCA (Lowest Common Ancestor), which is an image containing all relevant areas. This results in extremely high-resolution inputs. In contrast, our ***RAP*** employs a *Spatial-Awareness Layout* to selectively retain only the key image crops, effectively reducing image resolution while maintaining their spatial relationships (see Figure 4 in our paper).
>
> - **Search-based methods typically follow a top-down strategy, while *RAP* starts directly from low-resolution image crops**
>
>   Search-based methods often treat the high-resolution image as the root node, splitting it into sub-images (such as a $2\times 2$ grid) to generate child nodes. However, high-resolution root images can mislead the MLLM by providing incorrect search cues at the outset, resulting in inefficient paths and potential errors. Our ***RAP*** mitigates this issue by starting directly with low-resolution image crops, enabling more efficient and accurate retrieval of relevant content (see Table 5 in our paper).
>
> > **Q4:** Essential References Not Discussed.
>
> **R:** Thanks for pointing out the work on SV-RAG and MMed-RAG. SV-RAG applies RAG to multimodal long-document understanding, while MMed-RAG enhances the factuality of Med-MLLMs. In our revised version, we will include these two works in the discussion of Multimodal RAG in the Related Work section.

---

### Official Review · Reviewer_tave · 2025-03-12

**Overall Recommendation:** 4

**Summary:**

This paper introduces a retrieval-augmented perception method for MLLMs, which retrieve and fuses relevant image crops from the full high-resolution image.
Specifically, a apatial-awareness layout is proposed, which is to maintain the relative positional relationships of the image crops.
In addition, a retrieved-exploration search dynamically selects the optimal number of crops based on the text-to-crop similarity and the model's confidence.
In experiments, the proposed method outperforms others by a large margin in both fine-grained single-instance perception and fine-grained cross-instance perception tasks.

**Claims And Evidence:**

In the spatial-awareness layout, the compressed mapping is reversable so the proposed method can handle cross-instance perception task.

The paper appears to have successfully addressed the computational complexity limitations of search-based methods by progressively narrow down the search space with the proposed RE-Search inspired by A* algorithm rather than a simple divide and conquer approach or a complex search algorithm, in MLLMs' HR image perceptual capabilities.

**Essential References Not Discussed:**

.

**Experimental Designs Or Analyses:**

I'm uncertain whether the comparison methods are comprehensive enough, because I am not familiar with the HR image preception in MLLMs, but the types and quantities of comparison targets appear to be suitable, according to the `Related Work` section written by the authors.

The experimental results in the appendix effectively complement the experimental results presented in the main text.

However, the effect of crop size is not shown.

**Methods And Evaluation Criteria:**

The proposed method is evaluated on V∗ Bench and HR-Bench, showing improved performance by large margin with efficiency.

**Other Comments Or Suggestions:**

An analysis of failure cases is missing.
It is difficult to pick a specific case, but like all other methods the proposed method also have its pros and cons so it may have some shortcomings compared to existing methods.
Analyzing such cases will help for readers a deeper understanding of the proposed method.

**Other Strengths And Weaknesses:**

Section3 `Pilot Study` helps readers understand the intuition behind the proposed method more easily.
For example, Table1 shows that preserving relative position information is necessary to improve both FSP and FCP tasks.

**Questions For Authors:**

.

**Relation To Broader Scientific Literature:**

I believe this research could provide important contributions not only for HR image processing but also for extending into video perception.

**Theoretical Claims:**

The proposed RE-Search consists of two reward functions g and h inspired from A* search seems reasonable.
- g: similarity between current patches and the query.
- h: MLLM's confidence that the model can answer from given patches.

---

> ### Author Rebuttal · Authors · 2025-04-01
>
> We truly appreciate Reviewer tave's constructive comments and positive support.
>
> > **Q1:** The concern of the comparison methods are comprehensive enough.
>
> **R**: We'd like to thank the reviewer for the advice. In fact, building on established works such as $DC^2$ [R1] and ZoomEye [R2], we conduct experiments on $V^*$, HR-Bench, and the widely used benchmark MME-RealWorld. Our study includes a comprehensive comparison with **12 widely adopted MLLMs**, and we further demonstrate the efficiency of our *RAP* method against leading state-of-the-art approaches (see Table 5 in our paper).
>
> To echo the reviewer's concern, here we further strengthened our contributions by including additional experiments:
>
> - We kindly refer the reviewer to our response to `Reviewer h1HT (Q2)` for additional experimental results on **five widely used benchmarks**: DocVQA, ChartQA, TextVQA, AI2D, and MMStar.
> - We respectfully refer the reviewer to our response to `Reviewer TfPd (Q2)` for additional evaluations involving **five state-of-the-art MLLMs**, including LLaVA-ov-7B/72B, Oryx-1.5-7B, Cambrian-8B, and CogVLM-Llama3-19B.
>
> [R1] Wang W, Ding L, Zeng M, et al. Divide, conquer and combine: A training-free framework for high-resolution image perception in multimodal large language models[C]. In AAAI 2025.
>
> [R2] Shen H, Zhao K, Zhao T, et al. ZoomEye: Enhancing Multimodal LLMs with Human-Like Zooming Capabilities through Tree-Based Image Exploration[J]. arXiv preprint arXiv:2411.16044, 2024.
>
>
>
> > **Q2:** The effect of crop size.
>
> **R:**  We appreciate the reviewer for the valuable advice. We'd like to clarify that we use the default crop size of $448\times 448$ recommended by VisRAG in our *RAP*. To further investigate the impact of crop size, here we perform additional experiments on HR-Bench 8K using LLaVA-ov-0.5B and show the results below. Our findings reveal that while variations in crop size result in relatively minor differences, all configurations of our *RAP* method yield substantial performance gains over the baseline.
>
> |                             | HR-Bench 8K |
> | --------------------------- | ----------- |
> | LLaVA-ov-0.5B               | 42.8        |
> | LLaVA-ov-0.5B w/ *RAP* ($224\times 224$)    | 61.4        |
> | **LLaVA-ov-0.5B w/ *RAP* ($448\times448$)** | **63.5**    |
> | LLaVA-ov-0.5B w/ *RAP* ($896 \times 896$)   | 60.0        |
>
>
>
> > **Q3:** An analysis of failure cases is missing.
>
> **R:** We'd like to thank the reviewer for pointing out this issue. Admittedly, while *RAP* demonstrates strong capability in accurately retrieving key visual information, its performance can degrade when the input question lacks critical object-specific details. A detailed failure case is available at the following anonymous link:
>
> https://anonymous.4open.science/r/RAP_Case-1D48/Failure_Case_Study.md
>
> In our revised version, we will incorporate a comprehensive analysis of this case to further clarify the limitations and potential directions for improvement.

---

> > ### Comment · Reviewer_tave · 2025-04-07
> >
> > Thanks for addressing my questions, I keep my rating as accept.

---

### Official Review · Reviewer_h1HT · 2025-03-13

**Overall Recommendation:** 5

**Summary:**

The paper works on a key challenge in the area of MLLMs -- the perception of high-resolution (HR) images. Centering on this significant problem, this paper leverages RAG to enhance MLLM’s ability to perceive HR images. The paper first explores the impact of the layout and the number of retrieved image crops on the model’s perception capability and further proposes Retrieval-Augmented Perception (RAP) to improve MLLM’s perception of HR images. RAP retrieves image crops while preserving spatial relationships and dynamically select the optimal number of retrieved image crops via Retrieved-Exploration Search (RE-Search). To the best of my knowledge, this is the first study towards exploiting the benefits of visual RAG for the challenging HR perception tasks. Experiments achieve an improvement of 24% on average on HR benchmarks.

**Claims And Evidence:**

It looks to me that the paper mainly presents the following key claims:

1.	This paper finds that applying RAG to MLLMs for HR image perception requires preserving the original spatial information of the retrieved image crops and that the number of retrieved image crops needed varies across different tasks. The specific experimental support is provided in Section 3 named Pilot Study.

2.	Based on the above findings, this paper proposes a training-free framework, RAP, which maintains the relative positional relationships between retrieved image crops through Spatial-Awareness Layout and dynamically selects the appropriate top-K image crops using RE-Search. The authors show plenty of results in Tables 2 and 3, which convincingly show that RAP significantly enhances MLLM’s perception of hr images.

**Essential References Not Discussed:**

N/A

**Experimental Designs Or Analyses:**

The experiments validate the effectiveness of the proposed RAP method on both HR-Bench and MME-RealWorld. Although the authors claim to have evaluated on general benchmarks, the average resolution of MME-RealWorld is 2000×1500, which still falls under high resolution. It remains unclear whether the method is effective on more general MLLM benchmarks (e.g., MMStar).

**Methods And Evaluation Criteria:**

The proposed method is suitable for the task. The experiments conducted on the HR-Bench and MME-RealWorld are reasonable, as to my knowledge, many SOTA related works in this area also used the same benchmarks to evaluate the effectiveness of their methods.

[1] Wang et al. Divide, conquer and combine: A training-free framework for high-resolution image perception in multimodal large language models. 2024.

[2] Shen et al. ZoomEye: Enhancing Multimodal LLMs with Human-Like Zooming Capabilities through Tree-Based Image Exploration. 2024.

**Other Comments Or Suggestions:**

Please address:

1. Alternative computation of the confidence scores in RE-Search?

2. More experiments on more common resolution to show its generalization capability. I may consider raising my score, if the authors could show the effectiveness of their method and provide additional experiments on general benchmarks like MMStar.

**Other Strengths And Weaknesses:**

Strengths:

1. The paper is well-structured and clearly explains the motivation and insights in an intuitive manner, tackling a highly significant problem. The authors start with a pilot study, proposing three questions and challenges, which then motivate the main methodology design. This logic makes it easy to grasp its main idea.
2. The authors introduce RAP, which achieves superior performance on high-resolution benchmarks and is designed with the merits of easy integration into existing open-source MLLMs. I believe that this could potentially be of interest to a broad multimodal audience.

Weaknesses:

1. It looks to me that RAP relies on the model’s confidence scores in RE-Search, making it seemingly unsuitable for closed-source models. Are there other solutions that can replace the calculation of confidence scores in the RE-Search? Any more discussions on the alternative solution would be appreciated.

2. Although the authors claim to have validated their method on general benchmarks for standard MLLM scenarios, they only conducted experiments on the general benchmark MME-RealWorld, which has an average resolution of 2000×1500. In fact, it should be noted that some existing related studies (e.g. (Wang et al., 2024b) in the paper ) not only conduct experiments on HR benchmarks but also evaluate on benchmarks with normal resolution images, thus validating the generalization of the method. There is, however, a lack of experimental evidence demonstrating its applicability to more common resolutions (e.g., MMStar).

**Questions For Authors:**

One more minor question is in Table 1, I wonder why the FCP task still performs worse than the baseline even after preserving the spatial relationships of image crops?

**Relation To Broader Scientific Literature:**

RAP provides a novel and efficient method for the HR image perception of MLLMs. The method leverages the core idea of RAG to enhance the MLLM’s ability to perceive HR images.

**Theoretical Claims:**

N/A

---

> ### Author Rebuttal · Authors · 2025-04-01
>
> We truly appreciate Reviewer h1HT's insightful comments and suggestions.
>
> > **Q1:** Replace confidence score calculation in RE-Search.
>
> **R:** The reviewer's point is well taken. We clarify that currently, APIs like OpenAI's [R1] provide the `logprobs` parameter, which returns the log probabilities of output tokens and can be used as confidence scores in *RE-Search*.
>
> Admittedly, there are also some closed-source models that do not support `logprobs`. Here, to echo the reviewer's concern,  we explore a simple alternative approach using **generation-based confidence scores**. Specifically, we design a scoring prompt that asks the model to evaluate whether the given image contains sufficient information to answer the question. The constructed scoring prompt is as follows:
>
> ```Question: {}\nCould you answer the question based on the available visual information? Return only a JSON object with a numerical confidence score (0~10) of "Yes"  like {"Yes": x}.```
>
> In the implementation, we rescale the final score to the range $[0,1]$. We conduct additional experiments on HR-Bench using LLaVA-ov-0.5B. The experimental results are shown in the table below. Compared to the baseline, using *RAP* with generation-based confidence scores achieved an average improvement of 7.4%, demonstrating that generation-based confidence scores through the model can also lead to significant performance gains.
>
> The corresponding results and discussions will be included in the revision.
>
> |                                            | HR-Bench 4K | HR-Bench 8K |
> | ------------------------------------------ | ----------- | ----------- |
> | LLaVA-ov-0.5B                              | 51.5        | 42.8        |
> | **LLaVA-ov-0.5B w/ *RAP* (logit-based confidence score)**  | **61.3**    | **63.5**    |
> | LLaVA-ov-0.5B w/ *RAP* (generation-based confidence score) | 57.0        | 52.1        |
>
> [R1] OpenAI API, https://platform.openai.com/docs/api-reference/completions/create
>
> > **Q2:** Experimental results on general benchmarks.
>
> **R:** We appreciate the reviewer's advice. As suggested, we extend our *RAP* to LLaVA-v1.5 7B and 13B, evaluating it on DocVQA, ChartQA, TextVQA, AI2D, and MMStar. These five benchmarks cover resolutions ranging from 390 to 1600. Across both benchmarks and model scales, *RAP* consistently yields notable performance gains, reinforcing the robustness and generalizability of our approach. We will include the corresponding results on these broader benchmarks, along with further analysis, in the revised version.
>
> |  | DocVQA   | ChartQA  | TextVQA  | AI2D | MMStar |
> | -- | -- | -- | -- | -- | -- |
> | LLaVA-v1.5-7B  | 21.5     | 18.2     | 45.8 | 54.9     | 30.3     |
> | **LLaVA-v1.5-7B w/ *RAP***     | **26.2** | **18.5** | **46.8** | **55.1** | **33.1** |
> | LLaVA-v1.5-13B | 23.7 | 18.5 | 49.0 | 60.2 | 32.8  |
> | **LLaVA-v1.5-13B w/ *RAP***     | **26.0** | **23.2** | **50.0** | **60.9** | **34.4** |
>
>
> > **Q3:** Why does the FCP task still underperform despite preserving spatial relationships?
>
> **R:**  We thank the reviewer for the comments. We would like to clarify that the experiments in Table 1 were conducted using a fixed value of $K$. Our findings indicate that **the choice of $K$ significantly affects final performance**—using a fixed $K$ leads to suboptimal results on ***FCP*** compared to the baseline. To address this, we propose *RE-Search*, a method for adaptively selecting an appropriate $K$. As shown in Table 4, with *RE-Search* , both ***FSP*** and ***FCP*** outperform the baseline, demonstrating the effectiveness of our approach.

---

> > ### Comment · Reviewer_h1HT · 2025-04-04
> >
> > Thank the author for the response. Since the authors have addressed my main concern and, after reviewing the other reviewers’ comments, I am now inclined to increase my score for this paper. However, I still have a few minor concerns regarding the authors’ rebuttal. I believe further clarification on these points is essential to make the paper more solid.
> >
> > In the rebuttal, the authors proposed a generation-based confidence score, which requires the model to directly output its confidence in its own response. While this is a reasonable design—given that evaluation, benchmarking, and reflection based on large language models have become common practices and natural choices. However, generation-based confidence scores are ultimately just an alternative to logit-based confidence scores. They can be seen as an approximation of logit-based confidence scores. Therefore, before deciding to adopt generation-based confidence scores, it is necessary to analyze whether they can effectively approximate logit-based scores.
> >
> > 1. What is the degree of similarity between logit-based confidence scores and generation-based confidence scores?
> > 2. In addition, it would be helpful to provide concrete examples to intuitively illustrate: (a) what types of answers result in high generation-based confidence scores, (b) what types result in low scores.
> > 3. Generation-based confidence scores are also likely to be affected by prompt engineering. I suggest the authors present more alternatives for constructing such scores and conduct a comprehensive analysis of how different prompt designs impact the results.

---

> > > ### Author Response · Authors · 2025-04-07
> > >
> > > We truly appreciate Reviewer h1HT's insightful comments and suggestions, once again.
> > >
> > > > **Q1:** What is the degree of similarity between logit-based confidence scores and generation-based confidence scores?
> > >
> > > **R:** We analyze the correlation between generation-based and logit-based confidence scores on the HR-Bench 8K using the LLaVA-ov-0.5B. The findings are illustrated in the figure available at the following anonymous link:
> > >
> > > https://anonymous.4open.science/r/RAP_Case-1D48/confidence_experiment.png
> > >
> > > Our analysis reveals a high cosine similarity score of **0.97** between the two types of confidence scores, indicating a remarkable degree of alignment between their distributions. Interestingly, generation-based confidence scores tend to be consistently higher than their logit-based counterparts. This observation aligns with prior research [R1, R2], which suggests that LLMs, when used as evaluators, may introduce ***systematic biases***. Nonetheless, RAP utilizing generation-based confidence scores continues to deliver substantial improvements.
> > >
> > > [R1] Wu M, Aji A F. Style Over Substance: Evaluation Biases for Large Language Models[C] In COLING 2025.
> > >
> > > [R2] Wataoka K, Takahashi T, Ri R. Self-preference bias in llm-as-a-judge[J]. arXiv preprint arXiv:2410.21819, 2024.
> > >
> > > > **Q2:** It would be helpful to provide concrete examples to intuitively illustrate.
> > >
> > > **R:** We provide two examples to demonstrate (a) answers with low generation-based scores, and (b) those with high scores. A detailed figure is available at the following anonymous link:
> > >
> > > https://anonymous.4open.science/r/RAP_Case-1D48/confidence_illustrate.png
> > >
> > > Our findings indicate that when an image lacks useful information for answering the question, both generation-based and logit-based confidence scores tend to be relatively low. However, for images containing crucial information, images with higher resolution have lower logit-based confidence scores, while the generation-based confidence score remains higher.
> > >
> > > > **Q3:** Comprehensive analysis of how different prompt designs impact the results.
> > >
> > > **R:** To assess the impact of different prompt designs on generation-based confidence scores, we compare three distinct prompts:
> > >
> > > (1) A simple modification of the logit-based confidence score prompt, directly generating the confidence score:
> > >
> > >   ```
> > > Question: {}
> > > Could you answer the question based on the available visual information? Return only a JSON object with a numerical confidence_score (0~10) of "Yes" like {"Yes": x}.
> > >    ```
> > >
> > > (2) Expanding the confidence score range from $[0, 10]$ to $[0, 100]$:
> > >
> > >   ```
> > > Question: {}
> > > Could you answer the question based on the available visual information? Return only a JSON object with a numerical confidence_score (0~100) of "Yes" like {"Yes": x}.
> > >   ```
> > >
> > > (3) Providing more detailed descriptions, including the goal, scoring criteria for different score ranges, constraints on output format, and output examples:
> > >
> > >   ```
> > > # Goal
> > > Given a question: {} about visual content (e.g., images, charts, diagrams, or scenes), determine whether the question can be answered confidently using the available visual information. Return a JSON object with a numerical confidence_score (0-100) reflecting your certainty that the answer is "Yes." The confidence_score should be based on factors such as the clarity, relevance, and completeness of the visual information. For example:
> > >     - ​**​0-30​**​: Low confidence (e.g., visual information is missing, irrelevant, or too ambiguous).
> > >     - ​**​31-70​**​: Moderate confidence (e.g., partial or indirect visual evidence exists but requires assumptions).
> > >     - ​**​71-100​**​: High confidence (e.g., visual information directly and unambiguously answers the question).
> > >
> > > Ensure the response contains ​**​only​**​ the JSON object with the key "Yes" and the numerical score. Do not include explanations, markdown, or other text.
> > >
> > > # Output Format
> > > 1. JSON object with a single key "Yes" and an integer value between 0 and 100.
> > > 2. No additional keys or text allowed.
> > > 3. Score must reflect confidence in a "Yes" answer, even if the true answer is "No."
> > >
> > > # Example
> > > {"Yes": xx}
> > >
> > > # Your Answer
> > >   ```
> > >
> > > As shown in the table below, we observe that the prompt design has minimal impact on the final performance, while *RAP* consistently delivers significant improvements compared to the baseline (i.e., LLaVA-ov-0.5B). These results demonstrate that even the simplest prompt can yield strong performances, highlighting the robustness of our *RAP*.
> > >
> > > | | Prompt | HR-Bench 4K | HR-Bench 8K |
> > > | - | --| --| --|
> > > | LLaVA-ov-0.5B | - | 51.5 | 42.8 |
> > > | LLaVA-ov-0.5B w/ *RAP* (generation-based confidence score) | (1) | **57.0**  | **52.1**  |
> > > | LLaVA-ov-0.5B w/ *RAP* (generation-based confidence score) | (2) | 56.8 | 51.3 |
> > > | LLaVA-ov-0.5B w/ *RAP* (generation-based confidence score) | (3) | 55.4 | 51.3 |

---

### Decision · Program_Chairs · 2025-05-01

**Decision:**

Accept (oral)

**Comment:**

This paper addresses the challenge of enabling Multimodal Large Language Models (MLLMs) to effectively perceive high-resolution images. The authors propose Retrieval-Augmented Perception (RAP), a training-free framework that adapts principles from Retrieval-Augmented Generation (RAG). RAP operates by retrieving relevant image crops based on a textual query, fusing them while preserving spatial context via a proposed Spatial-Awareness Layout, and dynamically selecting the optimal number of crops using a Retrieved-Exploration Search (RE-Search) mechanism based on model confidence and retrieval scores.

Experimental results demonstrate substantial performance improvements on established high-resolution benchmarks (e.g., V* Bench, HR-Bench) when applying RAP to existing MLLMs. The paper effectively motivates the RAP design through initial pilot studies and presents the methodology clearly. The overall assessment indicates that RAP is a novel and practical framework offering a valuable contribution to high-resolution visual understanding in MLLMs.

Main Weaknesses: The RE-Search component relies on accessing model confidence scores (typically derived from logits), limiting its direct applicability to closed-source models that do not provide such access. Reviewers raised concerns regarding the initial scope of experiments, focusing primarily on specific high-resolution benchmarks and a limited set of base MLLMs. However, the authors provided substantial additional experiments during the rebuttal phase addressing these concerns by evaluating on broader MLLM benchmarks and more diverse, state-of-the-art models. A detailed comparison and differentiation from related search-based methods and native high-resolution MLLM architectures needed further clarification.

Overall, the strengths outweigh the weaknesses, and the reviewers are largely positive, so I recommend acceptance.

There are some clear areas for improvement though:
- Provide a thorough analysis and validation of the proposed generation-based confidence scoring mechanism as an alternative for closed-source models. This should include its correlation with logit-based scores and robustness across different prompt formulations.
- Incorporate the extensive supplementary experiments conducted during the rebuttal (evaluations on additional benchmarks like DocVQA/MMStar, results with models like Oryx/Cambrian/CogVLM, failure case analysis, crop size ablations, and refined efficiency metrics) into the main paper to fully demonstrate the method's robustness, generalizability, and practical trade-offs.
- It would also be interesting to check whether the retrieval is indeed enough to answer the questions (e.g., building on something like Sufficient Context: A New Lens on Retrieval Augmented Generation Systems, https://openreview.net/forum?id=Jjr2Odj8DJ). In the MLLM setting, this is still a very important consideration.